# Regulation of kinesin-2 motility by its β-hairpin motif

Stephanie Webb [1,4], Katerina Toropova [1,2,4], Aakash G. Mukhopadhyay [1,2], Stephanie D. Nofal [1,3] & Anthony J. Roberts [1,2] ✉

Members of the kinesin-2 family coordinate with other motors to power diverse physiological processes, but the structural mechanisms regulating kinesin-2 activity have been unknown. Distinctively, kinesin-2s canonically function as heterotrimers of two different motor subunits (for example Kif3A and Kif3B in humans) and Kap3, but the role of heterotrimerization has yet to fully emerge. Here, we combine structural, cell biological and single-molecule approaches to dissect kinesin-2 regulation as a heterodimer, heterotrimer and quaternary complex with a cargo adaptor (APC). We identify a conserved motif in the tail of kinesin-2s (the β-hairpin motif) that, in conjunction with the adjacent coiled coil, controls kinesin-2 motility by sequestering the motor domains away from their microtubule track. Our data reveal how Kap3 binds via a multipartite interface with Kif3A and Kif3B. Rather than activating motility directly, Kap3 provides a platform on which cargo adaptors can engage and occlude the β-hairpin motif. Together, these data articulate a structural framework for kinesin-2 activation, recycling by dynein and adaptation for different biological functions.

Understanding how motor proteins coordinate to spatially organize the cytoplasm is a frontier in structural cell biology. The kinesin-2 family, which predates the last eukaryotic common ancestor[1,2], lies at the heart of this phenomenon. Its members work in a coordinated fashion to transport cargo in diverse physiological processes[3,4], including axonal mRNA localization[5,6], vesicular trafficking[7–9], intraflagellar transport (IFT)[10–14], ciliogenesis and Hedgehog and Wnt signaling[15–17]. Loss-of-function mutations in mammalian kinesin-2 subunits are embryonically lethal, often with situs inversus[18], while recent studies in humans have identified mutations that are associated with ciliopathies[19,20]. Despite progress toward understanding the properties of kinesin-2 motor domains[21–25], the structural mechanisms of kinesin-2 coordination and motility regulation are poorly understood.

Distinctively among kinesins, members of the kinesin-2 family typically function as heterotrimers[26], in which two motor subunits (Kif3A and Kif3B or Kif3C in humans) heterodimerize and associate with a third subunit, Kap3, which is built around Armadillo (ARM) repeats[27–30]. In mammals, Kif3AB–Kap3 functions in IFT and cytoplasmic transport[3,31],

whereas Kif3AC–Kap3 is implicated in neuronal transport[32]. Each motor subunit contains a kinesin motor domain that binds microtubules and hydrolyzes ATP[33,34], coiled-coil segments that mediate heterodimerization[35–37] and a putatively disordered carboxy-terminal tail[28,29]. The binding site of Kap3 is controversial: one model posits that Kap3 binds to the coiled-coil domain[38,39], whereas others suggest that binding occurs through the disordered tail of one or both motor subunits[28,29,32,40]. Although essential for Kif3 function in vivo[13,41,42], Kap3's precise role is also unclear. Reconstitution studies of *Chlamydomonas reinhardtii* kinesin-2 indicate that the motor subunits can move along microtubules without Kap3, but their velocity is increased in its presence[23], whereas analysis of mouse proteins suggests that the kinesin-2 heterotrimer (Kif3AB–Kap3) is strongly autoinhibited[6]. Thus, the molecular basis and purpose of kinesin-2 heterotrimerization is yet to fully emerge.

Coordinated motor activity is particularly important in the elongated processes of neurons and within the confines of cilia and flagella[1]. In cilia, kinesin-2 powers the anterograde movement of IFT trains and

[1]Institute of Structural and Molecular Biology, Birkbeck, University of London, London, UK. [2]Sir William Dunn School of Pathology, University of Oxford, Oxford, UK. [3]Present address: The Francis Crick Institute, London, UK. [4]These authors contributed equally: Stephanie Webb, Katerina Toropova. ✉e-mail: anthony.roberts@path.ox.ac.uk

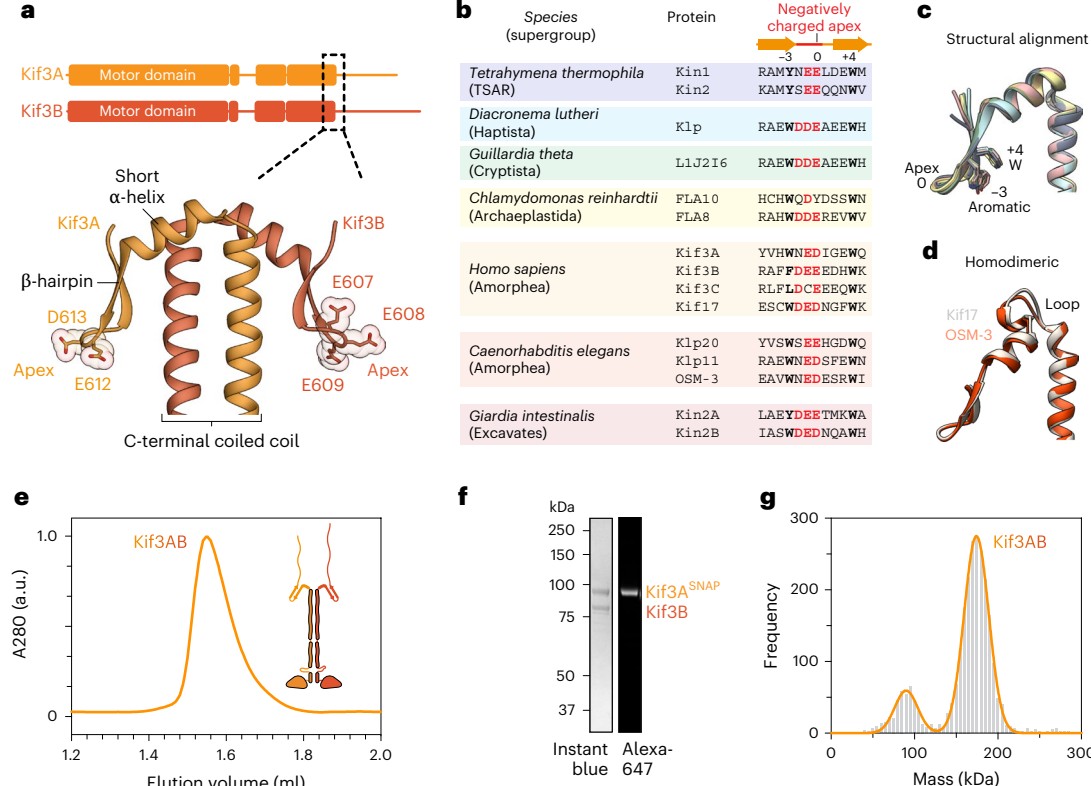

**Fig. 1 | A β-hairpin motif predicted in the tail of kinesin-2 in diverse eukaryotes.** **a**, Top, sequence diagrams of human kinesin-2 Kif3A and Kif3B. Bottom, AlphaFold3 model of the C-terminal coiled-coil region and β-hairpin motif, with the negatively charged residues at its apex shown in stick representation. The full AlphaFold3 model with pLDDT and PAE scores is shown in Extended Data Fig. 1. **b**, Structure-based sequence alignment of kinesin-2s from different eukaryotic supergroups. Negatively charged residues at the apex of the β-hairpin are shown in red. Conserved aromatic residues in the −3 and +4 positions relative to the apex are shown in bold. **c**, Aligned AlphaFold3 models of the β-hairpin in different kinesin-2s, colored as in **b**. **d**, AlphaFold3 models of homodimeric kinesin-2 Kif17 and OSM-3, which both feature an additional loop between the coiled coil and short α-helix preceding the β-hairpin. **e**, Size-exclusion chromatogram of reconstituted Kif3AB heterodimer with schematic of the construct inset. a.u., arbitrary units. **f**, SDS–PAGE of peak size-exclusion chromatography fraction after labeling SNAP-tagged Kif3A with the Alexa Fluor 647 fluorophore. **g**, Mass photometry of purified Kif3AB. The main peak (174 ± 15 kDa, mean ± s.d.) is consistent with the Kif3AB heterodimer mass (166 kDa).

cargoes needed for cilia assembly to the tip, before dynein-2 drives the return transport to the cell body[43,44]. To coordinate with other motors, kinesin-2 family members are thought to convert between autoinhibited and active states, with current models evoking folding of the molecule about a putative hinge in the coiled coil[3,21,23]. The molecular mechanism of motor autoinhibition is unclear, complicated by the question of whether one or both kinesin-2 motor subunits orchestrate the process. The issue is compounded by the fact that no autoinhibitory motif has been identified in the kinesin-2 subunits and the inhibitory sequence that regulates conventional kinesin-1 (the IAK motif[45]) is absent, suggesting that the regulatory mechanism may be novel. Valuable insights into kinesin-2 regulation have come from studies of homodimeric members of the family, Kif17 in humans and OSM-3 in *Caenorhabditis elegans*, which have demonstrated that tail truncations, mutations at the putative hinge site, surface attachment or binding of an associated factor (*C. elegans* DYF-1, also called IFT70) can stimulate motility[13,46–50]. Whether there is a unifying mechanism of regulation in the kinesin-2 family is unknown.

Here, we use cross-species structural analysis to identify a conserved motif in the tail of kinesin-2s (the β-hairpin motif), which was undetectable at the sequence level. We reveal that the β-hairpin motif mediates autoinhibition in kinesin-2, using negatively charged residues at the hairpin apex and the adjacent coiled coil to sequester the motor domains from their microtubule track. Our data show how Kif3 exploits heterotrimerization: Kap3 binds through a multipartite interface with Kif3A and Kif3B, creating a surface on which cargo adaptors can engage

and occlude the β-hairpin motif. Together, these data provide a structural basis for kinesin-2 activation, recycling in IFT and adaptation of family members for diverse functions.

## Results

### A β-hairpin motif predicted in the tail of kinesin-2 across eukaryotes

To investigate the basis for kinesin-2 autoregulation, we generated AlphaFold2 and AlphaFold3 models for a range of kinesin-2 sequences from across eukaryotic supergroups[51] (TSAR, Haptista, Cryptista, Archaeplastida, Amorphea and Excavates), including heteromeric motors (*Homo sapiens* Kif3A–Kif3B, *C. reinhardtii* FLA8–FLA10 and *C. elegans* KLP20–KL11) and homodimeric motors (*H. sapiens* Kif17 and *C. elegans* OSM-3). Notably, despite variations in the lengths of the coiled-coil segments and disordered tail region, a high-confidence feature consisting of a short α-helix and a β-hairpin (hereafter referred to as the β-hairpin motif) was found in all kinesin-2 models at the C-terminal end of the coiled coil (Fig. 1a and Extended Data Fig. 1). Structure-based alignment revealed that the apex of the β-hairpin invariantly contains negatively charged amino acids (Fig. 1b). Sequence constraints for residues flanking the apex are looser, consistent with the β-hairpin structure and indicating why the motif was not detected at the sequence level. However, there is strong conservation of aromatic residues in the +4 and −3 positions relative to the apex, which pack against each other (Fig. 1c). The invariance of the negatively charged apex across eukaryotic supergroups, in both heteromeric

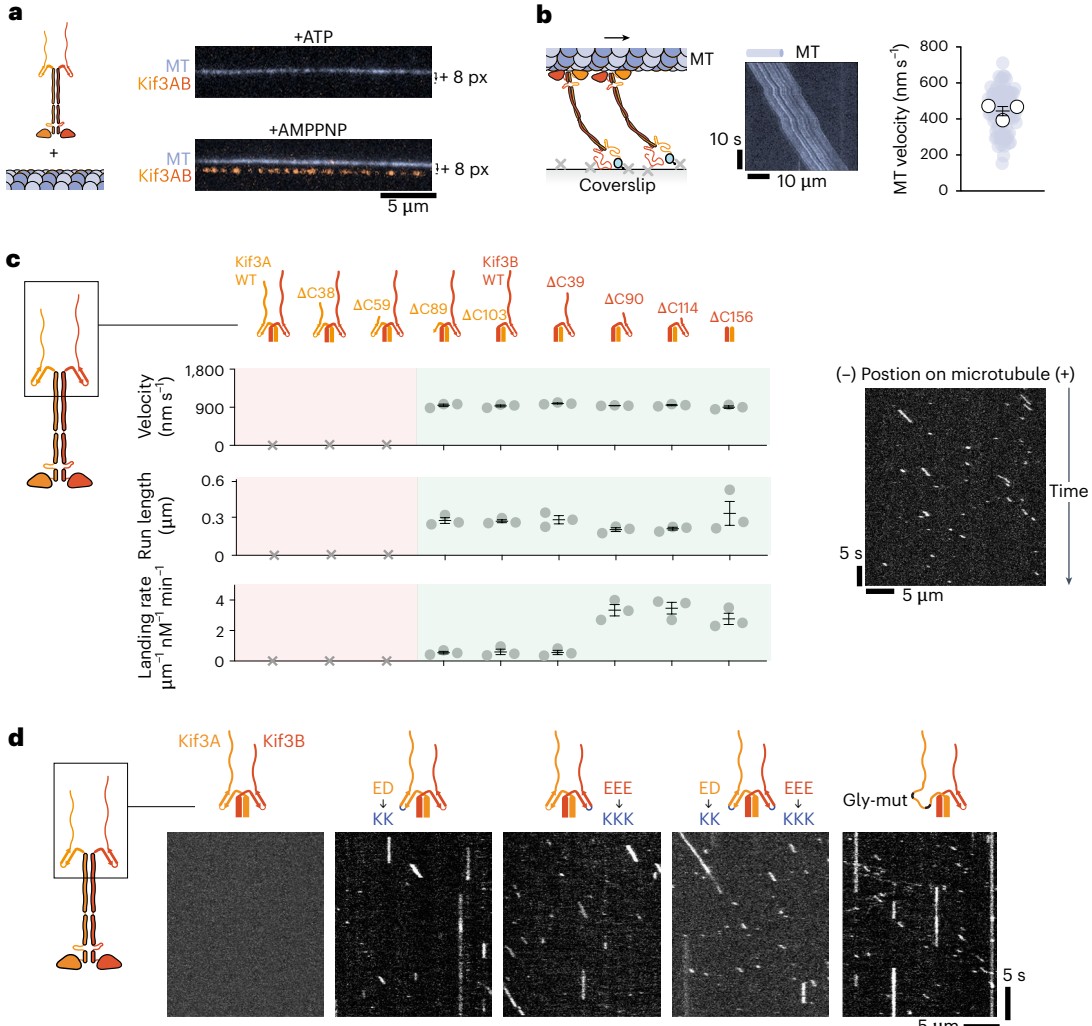

**Fig. 2 | The β-hairpin motif mediates kinesin-2 autoinhibition. a**, Composite TIRF images of the 488-nm channel (MT) and 640-nm channel (Alexa-Fluor-647-labeled Kif3A), offset in y axis by 8 pixels (px). Top, with 1 mM ATP. Bottom, with 1 mM AMPPNP. MT; microtubule. **b**, Left, schematic of the MT gliding assay with biotinylated Kif3AB. Middle, example kymograph. Right, plot of gliding velocities from three technical replicates. Colored circles; individual data points. White circles; average from each separate replicate. Lines, mean ± s.e.m., n = 33, 25 and 28 microtubules analyzed per replicate. **c**, Left, single-molecule velocity, run length and microtubule landing rate of Alexa-Fluor-647-labeled Kif3AB constructs on microtubules. Constructs were progressively truncated from their C termini, as indicated in the schematics above the plots. Measurements were taken from three technical replicates from separate motility chambers, with three fields of view per replicate. For velocity and run length, the numbers of measurements per technical replicate are: Kif3A$^{Δ89}$B, n = 120, 100 and 120; Kif3A$^{Δ103}$B, n = 113, 123 and 106; Kif3A$^{Δ103}$B$^{Δ39}$, n = 121, 105 and 102; Kif3A$^{Δ103}$B$^{Δ90}$,

n = 144, 108 and 103; Kif3A$^{Δ103}$B$^{Δ114}$, n = 105, 102 and 116; Kif3A$^{Δ103}$B$^{Δ156}$, n = 124, 108 and 104. For landing rate, n = 5 microtubules per replicate. Gray circles, average from each replicate. Lines, mean ± s.e.m. For Kif3A–Kif3B, Kif3A$^{ΔC38}$–Kif3B, and Kif3A$^{ΔC59}$–Kif3B constructs, no motile events were observed across three replicates (depicted with a cross), suggestive of strong autoinhibition at the single-molecule level. For other constructs, velocity and run-length values were not significantly different than each other (P > 0.1 one-way analysis of variance (ANOVA) followed by Tukey's multiple comparison test). Landing-rate values for Kif3A$^{Δ103}$B$^{Δ90}$, Kif3A$^{Δ103}$B$^{Δ114}$ and Kif3A$^{Δ103}$B$^{Δ156}$ were significantly higher than those of other constructs (P < 0.05). Exact P values are given in the Methods. Right, an example kymograph of Kif3A$^{ΔC103}$B$^{ΔC156}$. **d**, Example kymographs of Kif3AB β-hairpin mutants. Kif3A$^{KK}$ mutant: E612K D613K. Kif3B$^{KKK}$ mutant: E607K E608K E609K. Kif3A$^{Gly-mut}$: H609G W610G W617G Q618G. Motility parameters for all constructs are given in Extended Data Table 1.

and homodimeric kinesin-2s (Fig. 1c,d), suggests that it is ancestral, leading us to hypothesize that it plays an important role in kinesin-2 mechanism.

**Disruption of the β-hairpin motif activates kinesin-2 motility**
To investigate the role of the β-hairpin motif, we coexpressed and purified *H. sapiens* Kif3AB from insect cells. A SNAP$_f$ tag on Kif3A enabled covalent labeling with biotin (for surface immobilization in multi-motor microtubule gliding assays) or bright fluorophores (for single-molecule motility assays). Kif3AB was separated from excess ligand using size-exclusion chromatography (Fig. 1e), yielding purified (Fig. 1f) heterodimeric (Fig. 1g) protein.

In single-molecule motility assays, full-length Kif3AB did not detectably bind to or move along microtubules in the presence of ATP, suggestive of strong autoinhibition (Fig. 2a, top). In the presence of AMPPNP, which traps the kinesin motor domain in a high-microtubule-affinity state[52], full-length Kif3AB bound statically to microtubules (Fig. 2a, bottom). To test whether the lack of motility and microtubule binding in ATP conditions is due to autoinhibition at the single-molecule level, we biotinylated full-length Kif3AB and attached multiple motors to a neutravidin-coated surface through their tail (Fig. 2b). When microtubules and ATP were added, surface-immobilized Kif3AB molecules powered robust microtubule gliding at a speed of 444 ± 26 nm s$^{-1}$ (mean ± s.e.m. from 3 separate experiments). These

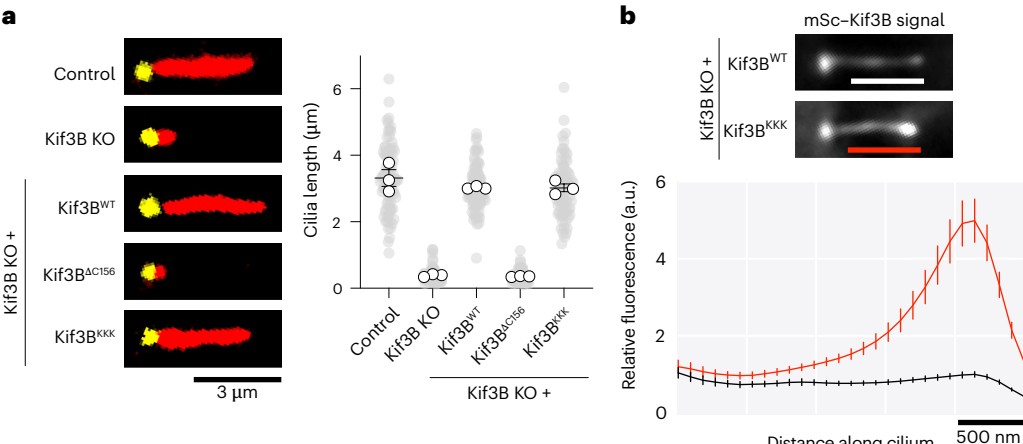

**Fig. 3 | The β-hairpin motif is crucial for Kif3 recycling and spatial regulation in intraflagellar transport. a**, Left, representative IMCD-3 cilia in control cells and Kif3B KO cells stably expressing the indicated constructs. Cells were immunofluorescently labeled for acetylated tubulin (red) marking the ciliary axoneme, and gamma tubulin (yellow), marking the basal body. Right, quantification of cilia length from three technical replicates. Gray circles, individual data points. White circles, average from each replicate. Lines, mean ± s.e.m. Control, $n = 86$; Kif3B KO, $n = 76$; Kif3B$^{WT}$, $n = 78$; Kif3B$^{ΔC156}$, $n = 76$; Kif3B$^{KKK}$, $n = 74$ cilia. Cilia length is not significantly different for control, Kif3B$^{WT}$ and Kif3B$^{KKK}$ cells ($P > 0.4$, one-way ANOVA followed by Tukey's multiple comparison test) or for Kif3B KO and Kif3B$^{ΔC156}$ cells ($P > 0.9$). Cilia length is significantly shorter in Kif3B KO and Kif3B$^{ΔC156}$ cells than in control, Kif3B$^{WT}$ or Kif3B$^{KKK}$ cells ($P < 0.0001$). Exact $P$ values are given in the Methods. **b**, Top, representative images of mScarlet (mSc)-tagged Kif3B$^{WT}$ and Kif3B$^{KKK}$ constructs expressed in Kif3B KO cells. Bottom, plot of the average mSc-Kif3B fluorescent signal from line scans along the cilium length (bars in the top panel indicate the distance analyzed), aligned at the ciliary tip (peak in the Kif3B$^{WT}$ trace). Values are normalized to Kif3B$^{WT}$ peak value. Traces show mean intensity ± s.e.m.; $n = 42$ (Kif3B$^{WT}$), 51 (Kif3B$^{KKK}$) cilia measured in three separate experiments. An analysis of Kap3 localization is shown in Extended Data Fig. 2e.

data indicate that full-length human Kif3AB can drive motility when multiple motors are anchored through their tail but is strongly auto-inhibited as a single molecule in solution.

To home in on the region or regions of Kif3AB that are responsible for autoinhibition, we made a series of C-terminal truncations (which did not affect heterodimerization; Extended Data Fig. 2a). Truncation of 38 or 59 putatively disordered amino acids from the C terminus of Kif3A had no effect on Kif3AB single-molecule motility, which remained strongly autoinhibited (Fig. 2c). Strikingly, however, truncations within or beyond the β-hairpin motif of Kif3A (89 and 103 amino acids, respectively), activated the motor, which bound to and moved along microtu-bules with an average velocity of ~940 nm s$^{-1}$, run length of ~0.3 μm and landing rate of ~0.6 μm$^{-1}$ nM$^{-1}$ min$^{-1}$ (Extended Data Table 1). These data indicate that disrupting or removing the Kif3A β-hairpin is sufficient to release Kif3AB autoinhibition. We next examined whether C-terminal truncations of Kif3B could further stimulate motility in constructs lacking the Kif3A β-hairpin (Fig. 2c). While truncation of 39 amino acids from the C terminus of Kif3B had no effect, removal of 90 amino acids increased the landing rate to ~3.2 μm$^{-1}$ nM$^{-1}$ min$^{-1}$, with velocity and run length remaining unchanged (Extended Data Table 1). Larger Kif3B C-terminal truncations of 114 or 156 amino acids (the latter removing the Kif3B β-hairpin motif) did not elicit further increases in motility. These results indicate that a segment of the Kif3B tail can suppress the landing rate, while raising the question of whether disrupting one β-hairpin motif in Kif3AB is sufficient for activation.

We next directly investigated the contributions of the Kif3A and Kif3B β-hairpin motifs to autoinhibition. We mutated the conserved negatively charged amino acids at the β-hairpin apex to lysine in Kif3A and Kif3B, individually or in tandem, in the context of the full-length subunits (Fig. 2d). Mutation of the Kif3A β-hairpin activated Kif3AB motility, as did mutation of the Kif3B β-hairpin, yielding motors that moved with similar, yet slightly longer, run lengths (~0.6 μm) and lower velocity (~730 nm s$^{-1}$) than those of constructs activated by truncation (Fig. 2d and Extended Data Table 1). To investigate whether the nega-tively charged residues need to be in the context of an intact β-hairpin for autoinhibition, we substituted glycine, the residue with lowest β-strand propensity, into the strands flanking the negatively charged

residues in Kif3A (while leaving the negatively charged residues them-selves intact; Gly-mut, Fig. 2d). This construct was also activated (Fig. 2d and Extended Data Table 1), in contrast to wild-type Kif3AB, indicating that an intact β-hairpin motif is required for autoinhibition. On the basis of these data, we conclude that (1) the β-hairpin motif has a fundamen-tal role in Kif3AB autoinhibition and (2) disruption of one β-hairpin, in either Kif3A or Kif3B, is sufficient to destabilize the autoinhibited conformation of Kif3AB.

## Impact of disrupting the β-hairpin on Kif3 cellular function

We next examined the role of the β-hairpin motif in Kif3's cellular activity, using the function of Kif3 in ciliary transport in IMCD-3 cells, a mouse kidney cell line, as an exemplar. IMCD-3 cells form a primary cilium upon serum starvation (cilia length of 3.3 ± 0.2 μm, mean ± s.e.m.) (Fig. 3a), in which Kif3 powers outward movement of IFT trains, transporting cargoes to the tip, where it is then returned back to the cell body on dynein-2 powered trains[53] (or by diffusion in *C. reinhardtii*[54]). We used CRISPR–Cas9 to generate a knockout (KO) of Kif3B (Extended Data Fig. 2b), which ablated cilia formation (Fig. 3a and Extended Data Fig. 2c). Stable expression of wild-type Kif3B–mScarlet in the KO cells rescued cilia length to 3.02 ± 0.02 μm, confirming that the loss of cilia was due to Kif3B deletion. Live-cell total internal reflection fluorescence (TIRF) microscopy revealed that Kif3B–mScarlet signal localized at the ciliary base and approximately evenly along the length of cilia (Fig. 3b, top), as quantified in line scans of the time-averaged fluorescence intensity (Fig. 3b, bottom). A Kif3B construct unable to bind to IFT trains and impaired in Kap3 binding[40] (Kif3B$^{ΔC156}$) did not restore cilia length, as expected (Fig. 3a), despite being expressed at comparable levels to the wild-type construct (Extended Data Fig. 2d). The Kif3B β-hairpin mutant (Kif3B$^{KKK}$), which was also expressed at comparable levels to the wild-type construct, did rescue cilia length (Fig. 3a), suggesting that it can deliver building blocks to the tip of the cilium to support ciliogenesis. However, unlike the wild-type construct, the Kif3B β-hairpin mutant accumulated in bright puncta at the ciliary tip (Fig. 3b). We also found that Kap3 accumulated at the ciliary tip in the Kif3B β-hairpin mutant background (Extended Data Fig. 2e). These results indicate that the constitutively active Kif3B β-hairpin mutant

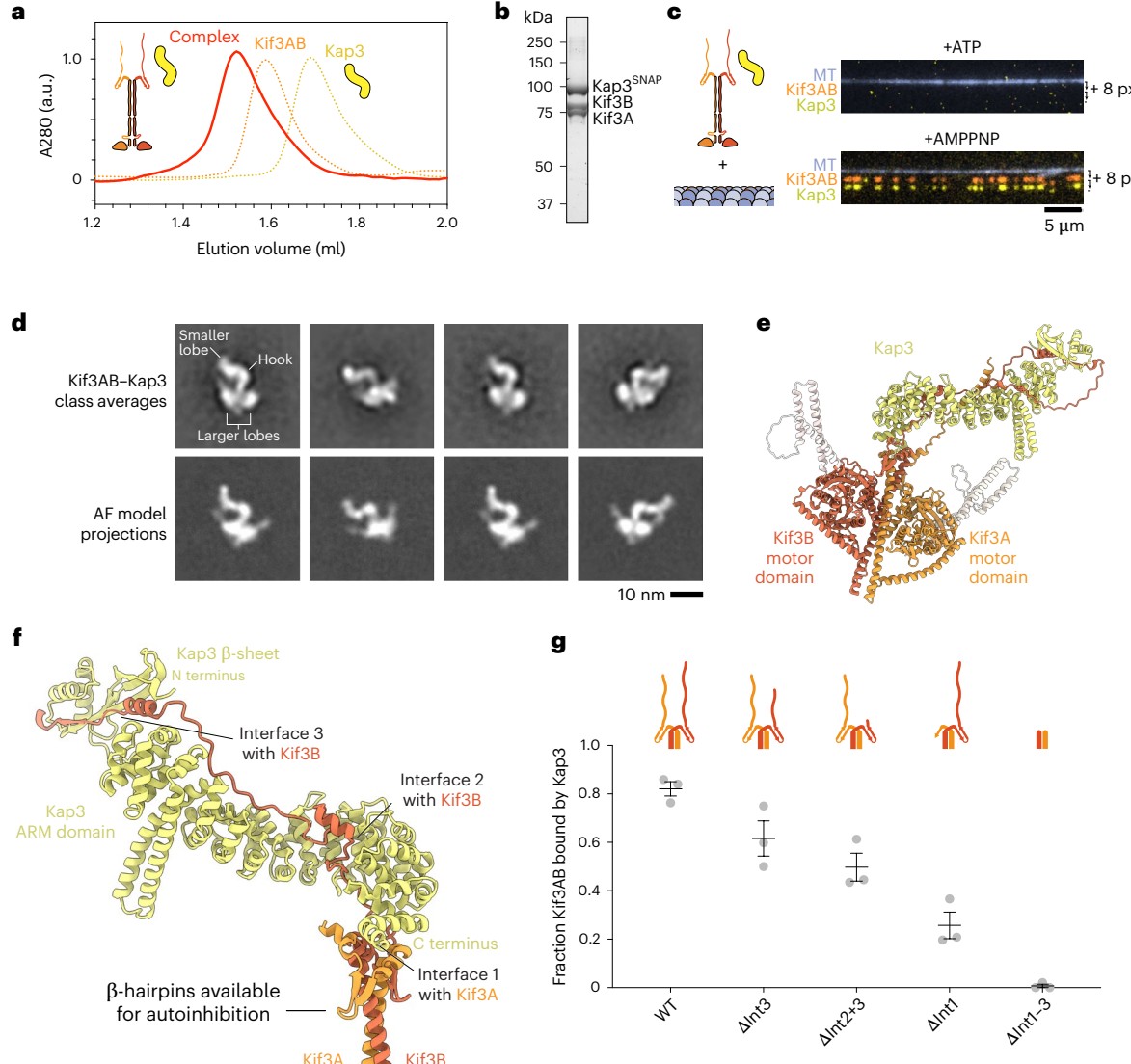

**Fig. 4 | Kap3 makes a multipartite interaction with Kif3AB that permits autoinhibition. a**, Size-exclusion chromatogram of reconstituted Kif3AB–Kap3 complex (red); schematics are also shown. Normalized Kif3AB and Kap3 traces are shown for comparison (dashed orange and yellow lines respectively). **b**, SDS–PAGE of the peak size-exclusion chromatography fraction. **c**, Top, composite TIRF image of the 488-nm channel (MT) and 640-nm channel (Alexa-Fluor-647-labeled Kif3A), offset in the *y* axis by 8 pixels, and the 561-nm channel (TMR-labeled-Kap3), offset in the *y* axis by 16 pixels. The assay was conducted with 1.5 nM Kif3AB, 7.5 nM Kap3 and 1 mM ATP. The presence of Kap3 does not activate Kif3AB microtubule binding or motility. Bottom, the same experiment as above, but with 1 mM AMPPNP. The assay was performed with 1 nM Kif3AB and 1 nM Kap3. Kif3AB and Kap3 colocalize in a complex. **d**, Top, example class averages of the Kif3AB–Kap3 complex, with features labeled. Bottom, corresponding AlphaFold3 (AF) model projections. **e**, Ribbon representation of the Kif3AB–Kap3 AlphaFold3

model. Protruding coiled-coil regions of Kif3AB shown in white. Further analysis of coiled-coil regions in this model shown in Extended Data Fig. 5. **f**, Close-up of the composite binding interface between Kap3 and Kif3AB C-terminal region in the AlphaFold3 model. The three main interfaces are labeled. Kif3AB β-hairpins are not occluded by Kap3 binding. **g**, Plot of the fraction of TMR-labeled Kap3 colocalizing with Alexa-Fluor-647-labeled Kif3AB for each indicated construct: ΔInt3 (Kif3A–Kif3B$^{\Delta C90}$), ΔInt2+3 (Kif3A–Kif3B$^{\Delta C114}$), ΔInt1 (Kif3A$^{\Delta C89}$–Kif3B) and ΔInt1–3 (Kif3A$^{\Delta C103}$–Kif3B$^{\Delta C156}$). Measurements were taken from three technical replicates. Gray circles, average from each replicate. Lines, mean ± s.e.m. WT, *n* = 49, 68 and 81; ΔInt3, *n* = 54, 75 and 85; ΔInt2+3, *n* = 46, 70 and 82; ΔInt1, *n* = 68, 44 and 56; ΔInt1–3, *n* = 59, 58 and 28 molecules analyzed per replicate. Mean values for WT, ΔInt2+3, ΔInt1 and ΔInt1–3 are significantly different from each other (*P* < 0.01, one-way ANOVA followed by Tukey's multiple comparison test). Exact *P* values are given in the Methods.

can power transport to the tip of cilia, to support ciliogenesis, but the Kif3 complex is not effectively returned to the cell body, highlighting autoinhibition through the β-hairpin motif as crucial for Kif3 recycling by dynein-2.

## Architecture of the Kif3 heterotrimer

We next sought to understand how Kif3AB binds to Kap3 to form the Kif3 heterotrimer. We purified human Kap3 from insect cells and reconstituted the heterotrimeric complex with full-length Kif3AB, which we verified through size-exclusion chromatography (Fig. 4a,b) and mass

photometry (Extended Data Fig. 3a). In single-molecule motility TIRF assays, Kif3AB–Kap3 did not bind to or move along microtubules in the presence of ATP, suggesting that Kap3 binds Kif3AB in a manner that maintains autoinhibition (Fig. 4c, top). In the presence of AMPPNP, Kif3AB and Kap3 bound statically to microtubules and colocalized with each other (Fig. 4c, bottom), showing that the complex is stable at the nanomolar concentrations of the TIRF assay.

To gain insight into how Kap3 binds to Kif3AB in a manner that allows autoregulation, we examined the structure of the heterotrimer using single-particle negative-stain electron microscopy (EM) (Fig. 4d

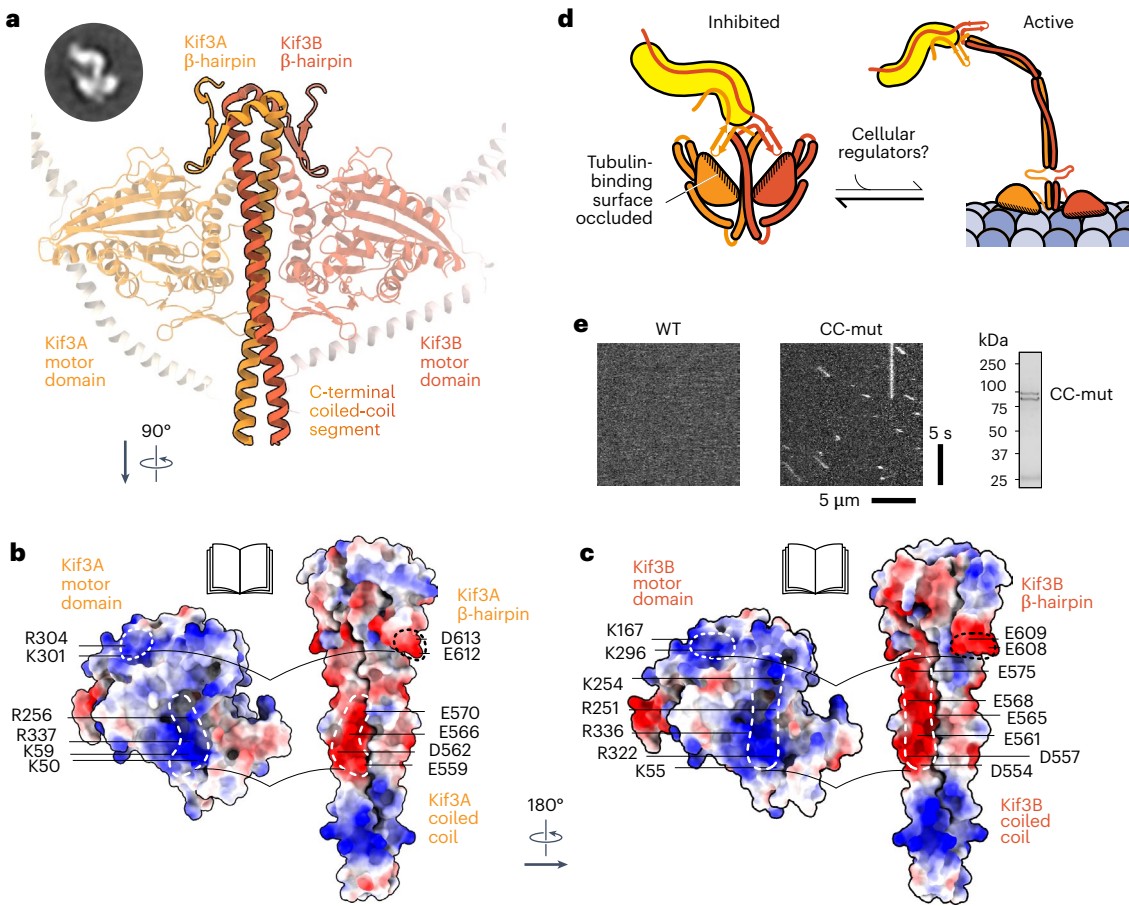

**Fig. 5 | Structural basis for Kif3 autoinhibition. a**, Close-up of the AlphaFold3 model of the autoinhibitory Kif3AB interface in cartoon representation, colored as in previous figures. Key features are labeled. The negative-stain EM class average from Fig. 4d is shown in the inset. **b**, Open-book representation of binding the interface between the Kif3A motor domain and Kif3A β-hairpin and coiled coil in the AlphaFold3 model. The model is shown in surface representation and colored by electrostatic potential with key residues annotated. Note the complementary charges between motor domain (positively charged; blue) and β-hairpin and coiled coil (negatively charged; red). **c**, Open-

book representation of the binding interface between Kif3B motor domain and Kif3B β-hairpin and coiled coil. **d**, Schematic of kinesin-2 autoinhibition. The Kif3AB motor domains fold back to interact with β-hairpin motifs and the C-terminal coiled coil, occluding the tubulin-binding surface of each motor. Kap3 binds C-terminal regions of Kif3AB and does not relieve autoinhibition, remaining available to bind other factors that might activate Kif3. **e**, Example kymographs of Kif3AB$^{WT}$ and Kif3AB$^{CC-mut}$ (Kif3A-E559K D562K E556K; Kif3B-D554K D557K E561K). The gels show purified Kif3AB$^{CC-mut}$. Motility parameters are provided in Extended Data Table 1.

and Extended Data Fig. 3b). Kif3AB–Kap3 displayed a compact shape comprising a hook-like density, matching the expected dimensions of the Kap3 ARM repeat α-solenoid[55], with a small lobe at one end and two larger lobes at the other. The larger lobes are consistent with the size of the kinesin motor domains, and a fine structure is occasionally visible projecting out from them. To help interpret these features and generate an atomic hypothesis for the binding mode, we made an AlphaFold3 model of the Kif3AB–Kap3 heterotrimer (Fig. 4e). Projections of this model showed striking concordance with the experimental EM class averages (Fig. 4d) and support assignment of the hook-like feature as Kap3 and the two larger lobes as the Kif3A and Kif3B motor domains (Fig. 4e).

In this high-confidence model, Kap3 binds to Kif3AB through multiple interfaces (Fig. 4f and Extended Data Fig. 3c). First, the C-terminal end of Kap3's ARM domain interacts with a pocket formed by short helices of Kif3A flanking the β-hairpin motif (interface 1). Notably, this contact leaves the apex of the β-hairpin motifs free to interact with the Kif3AB motor domains (described below), indicating why the Kif3 heterotrimer remains autoinhibited. Second, the disordered Kif3B tail runs along the inner surface of the Kap3 ARM α-solenoid (interface 2). Notably, this site encompasses a conserved phosphorylation site in Kif3B[56] (see Discussion). Third, the C-terminal region of the Kif3B tail

forms a β-strand and α-helix that interact with a three-stranded β-sheet at the Kap3 amino terminus (interface 3). This β-sheet corresponds to the smaller lobe at the distal end of Kap3 observed in our EM class averages. Finally, the disordered Kif3A tail also interacts with the Kap3 ARM domain, although less extensively than the Kif3B tail (Extended Data Fig. 3c).

To test the importance of these interfaces in Kap3 binding, we performed a sensitive single-molecule binding assay, in which Kif3AB and Kap3 were labeled with different color fluorophores (Fig. 4c,g). Whereas Kap3 colocalized with 82 ± 3% of full-length Kif3AB molecules (mean ± s.e.m.), ablation of interface 3 by truncation of the Kif3B tail reduced colocalization to 62 ± 7%, which was further reduced to 50 ± 6% by removal of interface 2. Removal of interface 1, by truncation of Kif3A, had a stronger effect, reducing colocalization to 26 ± 6%. Finally, removal of all the predicted interfaces, while retaining the N-terminal portion of Kif3AB, abolished colocalization, showing that Kap3 does not bind appreciably to the Kif3AB coiled-coil or motor domains. We also found that Kap3 binds to full-length Kif3AC, but not to a construct comprising the Kif3AC coiled-coil and motor domains (Extended Data Fig. 3d). In summary, we conclude that Kap3 makes a multipartite interaction with the C-terminal regions of both Kif3A (interface 1) and Kif3B (interfaces 2 and 3).

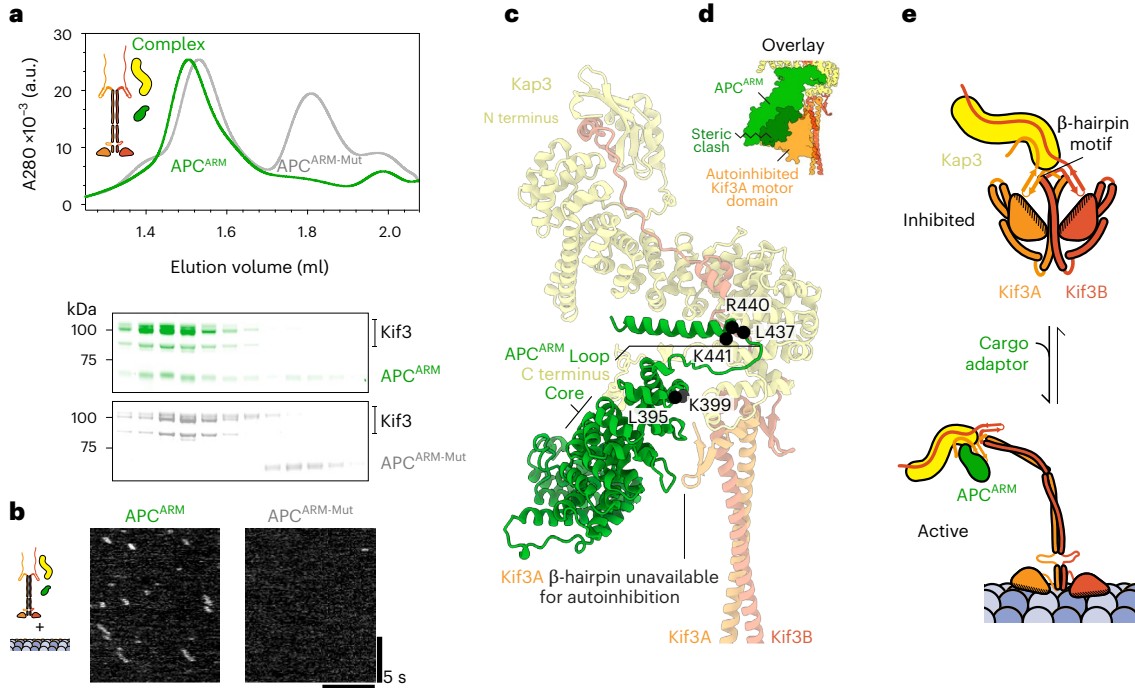

**Fig. 6 | Kap3-dependent activation of Kif3 by adaptor binding to the β-hairpin motif. a**, Top, size-exclusion chromatogram of reconstituted Kif3AB–Kap3 complex in the presence of either APC$^{ARM}$ (green) or APC$^{ARM-Mut}$ (L395Y K399E K441E L437Y R440E, gray). Bottom, SDS–PAGE of size-exclusion chromatography fractions. Traces for APC$^{ARM}$ and APC$^{ARM-Mut}$ alone shown in Extended Data Fig. 4a. **b**, Example kymographs of Kif3AB-Kap (Alexa-Fluor-647-labeled Kif3A) in the presence of microtubules and APC$^{ARM}$ or APC$^{ARM-Mut}$. **c**, AlphaFold3 model of APC$^{ARM}$ in complex with Kif3AB–Kap3. Conserved APC$^{ARM}$ residues at the interface are shown as black spheres and annotated (mutated in APC$^{ARM-Mut}$). APC$^{ARM}$

occludes the Kif3A β-hairpin. **d**, Equivalent view to **c**, with the autoinhibited Kif3A motor domain binding site overlaid, showing steric clash with APC$^{ARM}$. The APC$^{ARM}$ and Kif3A motor domain are shown in surface representation. **e**, Schematic model of kinesin-2 activation. Isolated Kif3 favors a compact autoinhibited conformation in which the β-hairpin motifs and C-terminal coiled coil interact with the motor domains, sequestering the motor domains from the microtubule. Binding of a cargo adaptor, exemplified here by APC$^{ARM}$, occludes the β-hairpin motif, promoting an extended activated conformation in which the motor domains are free to drive processive motility along the microtubule.

## Structural basis for Kif3 autoinhibition

Our EM class averages, mutagenesis data and AlphaFold analysis suggest a structural framework for Kif3 autoinhibition (Fig. 5). In this model, the Kif3A and Kif3B motor domains lie on either side of the C-terminal coiled-coil segment, interacting with (1) their respective β-hairpin motif and (2) the C-terminal Kif3AB coiled coil (Fig. 5a). Both interactions have a major electrostatic component and involve the same residues that the kinesin motor domain would otherwise use to bind the globular core of α- and β-tubulin. A basic patch in the Kif3A motor domain (involving Lys301 and Arg304) interacts with the acidic apex of the Kif3A β-hairpin (Glu612, Asp613), echoing the interaction with β-tubulin[57] (Fig. 5b). A second basic patch in the Kif3A motor domain (involving Lys50, Lys59, Arg256 and Arg337) interacts with a cluster of electronegative residues in the C-terminal coiled coil (Kif3A Glu559, Asp562, Glu566 and Glu570), echoing the interaction with α-tubulin[57] (Fig. 5b). The Kif3B motor domain shows similar, but non-identical, interactions (Fig. 5c). This suggests why chimeras in which the Kif3A motor domain is transplanted onto the Kif3B C-terminal region, and vice versa, are not autoinhibited[20,21]. Together, these interactions hold the motor domains of Kif3A and Kif3B in a pseudo-twofold symmetrical arrangement with their microtubule-binding domains occluded, consistent with the lack of Kif3AB microtubule binding observed in our motility assays (Fig. 5d). To test the importance of the interaction between the Kif3AB motor domains and C-terminal coiled-coil segment for autoinhibition, we mutated three negatively charged residues in the Kif3A coiled coil (E559K, D562K and E566K) and Kif3B coiled coil (D554K, D557K and E561K) at the interface with the autoinhibited motor domains (CC-mut). Although the mutations had no effect on Kif3AB heterodimerization, they activated the motility of Kif3AB, akin to the

removal of the β-hairpin motif (Fig. 5e and Extended Data Table 1). We conclude that both the β-hairpin motif and adjacent C-terminal coiled coil are critical for Kif3AB autoinhibition. Our data suggest that the autoinhibited conformation of Kif3AB is meta-stable, as AMPPNP or mutagenesis of only one β-hairpin is sufficient to disfavor it (Fig. 2), indicating that it could be a target for regulation.

## Activation of Kif3 by adaptor binding to the β-hairpin motif

How might cellular factors act on the subunits of Kif3 to elicit activation? To address this question, we purified adenomatous polyposis coli (APC), a well-characterized adaptor that couples Kif3 to mRNA cargoes in neurons and whose ARM domain (APC$^{ARM}$) is involved in Kif3 motility regulation[6,32,55,58]. Size-exclusion chromatography demonstrated that APC$^{ARM}$ binds stably to the Kif3AB–Kap3 heterotrimer (Fig. 6a), but weakly to Kif3AB alone (Extended Data Fig. 4a,b), suggesting that Kap3 forms an important part of the Kif3–APC$^{ARM}$ interface. Examination of Kif3AB–Kap3 in the presence of APC$^{ARM}$ by negative-stain EM showed more elongated, conformationally heterogeneous particles (Extended Data Fig. 4d) compared with the compact, autoinhibited conformation of Kif3AB–Kap3 alone (Fig. 4d and Extended Data Fig. 3b). Notably, single-molecule TIRF assays showed that APC$^{ARM}$ was sufficient to activate the motility of Kif3AB–Kap3, which bound to and moved along microtubules (Fig. 6b) with motile parameters akin to constructs activated by the β-hairpin mutation (Extended Data Table 1). In contrast, APC$^{ARM}$ did not activate the motility of Kif3AB alone, consistent with the binding assay (Extended Data Fig. 4a–c). Together, these data indicate that APC$^{ARM}$ activates Kif3 in a Kap3-dependent manner.

To explore the basis for Kif3 activation by APC$^{ARM}$, we generated an AlphaFold3 model of their interface (Fig. 6c). In this model, APC$^{ARM}$

binds a composite surface consisting of the C-terminal region of the Kap3 ARM domain and, strikingly, the β-hairpin motif of Kif3A, with each interaction supported by low predicted aligned error (Extended Data Fig. 4e). The interaction with the Kap3 ARM domain involves the core of APC[ARM] and a loop that protrudes from its C-terminal end. The disordered C-terminal region of Kap3 also contacts the internal surface of APC[ARM]. Notably, the binding footprint of APC[ARM] sterically clashes with the position of the Kif3A motor domain in the autoinhibited conformation of Kif3 and occludes the Kif3A β-hairpin (Fig. 6c,d), suggesting why APC[ARM] activates Kif3, as observed in our motility assays (Fig. 6b). The notion that occlusion of one β-hairpin is sufficient for activation is consistent with our data showing Kif3AB autoinhibition can be relieved by mutagenesis of a single β-hairpin (Fig. 2d). To test this activation model, we mutated conserved surface residues in APC[ARM] at the putative interface with Kif3 (Fig. 6c). Whereas wild-type APC[ARM] robustly coeluted with Kif3 in a size-exclusion chromatography assay, binding was effectively abolished by the interface mutations (Fig. 6a). Moreover, in contrast to wild-type APC[ARM] that activated Kif3 motility, the APC[ARM] interface mutant failed to relieve Kif3 autoinhibition in single-molecule TIRF assays (Fig. 6b), demonstrating that binding at this interface is the cause of Kif3 activation. Together, these results indicate that APC[ARM] activates Kif3 motility in a Kap3-dependent manner by binding to a composite interface comprising Kap3 and the Kif3A β-hairpin.

## Discussion

Here, using purified human proteins, EM, AlphaFold, single-molecule TIRF and CRISPR–Cas9 genome editing, we have dissected the mechanism of Kif3 motility regulation and heterotrimerization. Our results have implications across the kinesin-2 family: one of the 'toolbox' motors present in the last eukaryotic common ancestor, whose extant members function in IFT and cilia self-assembly, Hedgehog and Wnt signaling, vesicle transport and axonal mRNA localization, among other vital physiological roles[1–3,15–17].

We find that a conserved β-hairpin motif mediates autoinhibition in kinesin-2 by using negatively charged residues at the hairpin apex and adjacent C-terminal coiled coil. The β-hairpin and coiled coil bind to the motor domains through their α- and β-tubulin interacting surface and sequesters them from their microtubule track. This structural mechanism, derived from our EM data, pseudo-atomic model and mutagenesis, is compatible with foundational images of sea urchin kinesin-2 in a compact morphology captured by rotary shadow EM[28]. The ubiquity of the β-hairpin motif we find predicted in kinesin-2s across eukaryotic supergroups, and its absence in other kinesin families, suggests that it may serve as a useful tool for identifying kinesin-2 members, for example in poorly annotated proteomes in which motor-domain-based classification is ambiguous. One rare exception is trypanosomatids, a group we have found to lack a predicted β-hairpin in their kinesin-2 sequences (which also display other divergent features[59]), suggesting that the regulatory mechanism in these human parasites may be unique.

The β-hairpin mechanism we describe provides a basis for why a C-terminal Kif3A fragment (which we can now see includes the β-hairpin motif) inhibits Kif3AB in trans[60], and why mutations at the putative hinge site in the coiled coil relieve kinesin-2 autoinhibition[20,21,46,48] (as they would disfavour access of the β-hairpins to the motor domains). However, our EM data and analysis of the coiled coil (Extended Data Fig. 5) suggest that formation of the inhibited state is more complex than folding of the molecule about the putative hinge. Rather, our data indicate that in the compact, inhibited state we observe, the coiled coil segments proximal to the motor domains separate from one another and pack against themselves to shield their hydrophobic seams, whereas formation of the active state would involve zippering of these α-helical segments together into a canonical coiled coil (Extended Data Fig. 5). This coiled coil separation and unique use of the β-hairpin

motif distinguishes the autoinhibited conformation of kinesin-2 from that of kinesin-1, in which the coiled coil segments pack against each other in a hierarchical pattern, while the two α-helices of the coiled coil remain in apposition[61,62].

Our data indicate that heterotrimeric kinesin-2 intrinsically exists in a compact, β-hairpin-engaged state, but transiently samples the extended (β-hairpin-disengaged) state, consistent with sedimentation analysis[28] and providing a plausible explanation for previous observations. For example, the finding that *C. reinhardtii* FLA10-FLA8 can move along microtubules whereas mouse and human Kif3AB are strongly autoinhibited can be interpreted by differing buffer conditions or phosphorylation affecting the equilibrium position between the states[6,23,63]. It is also of note that FLA10 has fewer negative charges at the apex of its β-hairpin compared to many of its Kif3A orthologs, which could favor the β-hairpin-disengaged state (Fig. 1b). In vivo, we think it likely that the β-hairpin-engaged state predominates for isolated kinesin-2, because β-hairpin-mediated autoinhibition provides the opportunity for controlled activation of motility. In line with this proposal, Kif3AB and homodimeric kinesin-2, Kif17, exist in an autoinhibited state when expressed in cells[20,48].

Our data suggest how kinesin-2 exploits heterotrimerization: Kap3 binds through a multipartite interface with Kif3A and Kif3B, and—rather than activating motility directly—provides a surface on which cargo adaptors, exemplified by APC in this study, can stably engage and occlude the β-hairpin motif. APC activates kinesin-2 for mRNA transport in neurons[6], and we anticipate that other adaptors activate kinesin-2 in different biological contexts using analogous mechanisms. For example, binding of heterotrimeric kinesin-2 to the IFT-B complex may contribute to the initiation of anterograde IFT[64]. In *C. elegans*, in which homodimeric kinesin-2 OSM-3 drives anterograde transport along the distal ciliary segment[13,14,65], binding of OSM-5, DYF-1, DYF-6 and OSM-6 (also known as IFT88, IFT70, IFT46 and IFT52) to the extended C-terminal region of OSM-3 could occlude the β-hairpin motif and explain the observed motor activation[49]. It will be interesting to examine whether distinct adaptors underlie Kif3-mediated transport of other cargoes, a striking example being the regulated transport of melanosomes that underlies the color changes of amphibians[9].

We show that, rather than binding to the coiled coil or motor domains, Kap3 engages Kif3AB through multiple interfaces in their C-terminal regions. Among the interfaces between Kap3 and Kif3AB we describe, interfaces 2 and 3, involving the Kif3B tail, are particularly interesting from a regulatory perspective. Interface 2 includes a conserved phosphorylation site, which in *C. reinhardtii* influences IFT turnaround[56]. Interface 3 is a main point of departure between Kif3AB–Kap3, which functions in IFT, and Kif3AC–Kap3, which does not[31]. We find that Kif3AC is predicted to interact with Kap3 using interfaces 1 and 2, consistent with binding experiments[32] (Extended Data Fig. 3d), but lacks interface 3 (Extended Data Fig. 3c). Thus, interface 3, which involves the C-terminal region of Kif3B contributing a β-strand and α-helix to a β-sheet at the Kap3 amino terminus, might dictate kinesin-2 specificity for IFT.

Members of the kinesin-2 family coordinate with other motors to power diverse physiological processes[3]. Our cellular experiments show that β-hairpin-mediated autoinhibition is critical for the spatial regulation of Kif3 in IFT, as the motor accumulates at the ciliary tip when this mechanism is disabled and cannot be effectively returned to the cell body by dynein-2. We foresee that β-hairpin-mediated autoinhibition will be used widely in the kinesin-2 family to facilitate coordination with other motors, a hypothesis that can be tested using the β-hairpin mutations and structural framework for kinesin-2 regulation established here.

## Online content

Any methods, additional references, Nature Portfolio reporting summaries, source data, extended data, supplementary information, acknowledgements, peer review information; details of author contributions

and competing interests; and statements of data and code availability are available at https://doi.org/10.1038/s41594-025-01630-5.

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

## Methods

### Expression and purification of Kif3 and APC constructs

Genes encoding human Kif3A, Kif3B, Kif3C, Kap3 and APC$^{ARM}$ (residues 316–779) were synthesized (Epoch, Eurofins or IDT) and inserted into the pACEBac1 vector (Geneva Biotech) using HiFi assembly (NEB) with the following sequences added: Kif3A, C-terminal SNAP$_f$ tag, TEV protease site and FLAG tag or C-terminal TEV protease site and FLAG tag; Kif3B and Kif3C (Kif3B/C), N-terminal ZZ tag and TEV protease site; Kap3 and APC$^{ARM}$, N-terminal ZZ tag, TEV protease site and SNAP$_f$ tag. The wild-type sequences were altered to generate deletions and mutations using Q5 site-directed mutagenesis reactions (NEB). All constructs were verified by DNA sequencing.

Constructs were expressed in *Spodoptera frugiperda* (Sf9) cells (Thermo Fisher Scientific) using the baculovirus system. For Kif3AB and Kif3AC expression, 250-ml cultures were coinfected with Kif3A and Kif3B/C V$_1$ viruses at 2%:0.5% Kif3A:Kif3B/C (vol/vol). Protein purifications were performed at 4 °C, as described previously[66,67] and repeated here for completeness. Frozen cell pellets, typically from 250 ml of Sf9 culture, were resuspended in 20 ml purification buffer (30 mM HEPES pH 7.4, 300 mM KCl, 50 mM K-acetate, 2 mM Mg-acetate, 1 mM EGTA, 10% (vol/vol) glycerol, 1 mM DTT, 0.2 mM Mg-ATP, 1 mM PMSF) supplemented with a cOmplete EDTA-free Protease Inhibitor Cocktail (Roche). Cells were lysed using a Dounce homogenizer with 10 strokes of a small clearance pestle. Lysates were clarified by ultracentrifugation in a Type 70 Ti rotor at 183,960$g$ for 30 min. The supernatant was incubated for 1.5 h with 0.5 ml IgG Sepharose 6 resin and gentle rolling (GE Healthcare). Resin and bound proteins were collected by gentle centrifugation at 670$g$ for 5 min, transferred into a 20-ml column and washed with 2 × 20 ml volumes of purification buffer and 1 × 10 ml volume of TEV buffer (50 mM Tris pH 7.5, 150 mM K-acetate, 2 mM Mg-acetate, 1 mM EGTA, 10% (vol/vol) glycerol, 1 mM DTT, 0.2 mM Mg-ATP). Proteins of interest were eluted by resuspending the resin in TEV buffer, adding 100 µg TEV protease and incubating the reaction overnight on a roller. TEV-cleaved proteins were separated from the resin using an empty column and aliquots flash frozen in liquid nitrogen. Proteins were labeled with SNAP-Surface Alexa Fluor 647, SNAP-Cell TMR-Star or SNAP-Biotin (NEB) by incubating 5–10 µM proteins with a threefold molar excess of SNAP-ligand on ice for 1.5 h in a 100-µl reaction volume. Proteins were then cleared by ultracentrifugation in a TLA 100 rotor at 386,400$g$ for 10 min and purified from excess SNAP ligand by size-exclusion chromatography, as described below.

### Size-exclusion chromatography and complex reconstitution

Proteins (5–10 µM in 100 µl) were analyzed using size-exclusion chromatography using either ÄKTAmicro or ÄKTA Go systems with a Superose 6 Increase 3.2/300 column (Cytiva) in buffer A (50 mM Tris-HCl pH 7.5, 150 mM K-acetate, 2 mM Mg-acetate, 5% glycerol, 1 mM DTT, 0.2 mM Mg-ATP). For reconstitution of Kif3AB–Kap3, Kif3AB and Kap3 were incubated on ice at a 1:1 molar ratio for 15 min. For reconstitution of Kif3AB–Kap3–APC$^{ARM}$, the pre-formed Kif3AB–Kap3 complex was incubated for 10 min with a twofold molar excess of APC$^{ARM}$ or APC$^{ARM-Mut}$. Fractions of 50 µl were collected and analyzed by SDS–PAGE on 4–12% NuPAGE Bis-Tris gels (Thermo Fisher Scientific). For SNAP-labeling reactions, peak fractions were flash frozen as 5-µl aliquots in liquid nitrogen.

### Mass photometry

Kif3AB and Kap3 complexes were reconstituted following the protocol for size-exclusion chromatography experiments and then diluted to 1–100 nM using buffer A (described above). Mass photometry measurements were recorded using the Refeyn OneMP instrument[68] and analyzed using Discover software (Refeyn). NativeMark unstained protein standards (Thermo Fisher Scientific) were used to generate molecular mass calibration curves.

### Motility assays

Microtubules were polymerized from porcine tubulin (Cytoskeleton). A mixture comprising 10 mg ml$^{-1}$ tubulin, 1 mM GTP, 1 mM MgCl$_2$, 1 mM DTT and 10% DMSO was assembled in BRB80 (80 mM PIPES pH 6.9, 2 mM MgCl$_2$, 1 mM EGTA) on ice, then incubated at 37 °C for 30 min. Following the addition of an equal volume of BRB80 and 40 µM taxol, the solution was incubated for 10 min at 37 °C, then stored at ambient temperature. For fluorescent visualization or surface attachment, 10% of HiLyte 488 tubulin or biotin tubulin (Cytoskeleton) was included in the polymerization mixture, respectively.

Motility assays were carried out in flow chambers assembled between glass slides, biotin-PEG (Stratech) coverslips and double-sided tape. For microtubule gliding assays, flow chambers were sequentially incubated with (1) blocking solution (0.75% Pluronic F-127, 5 mg ml$^{-1}$ casein) for >5 min, followed by two washes with buffer B (50 mM Tris-HCl pH 8.0, 50 mM KCl, 2 mM MgCl$_2$, 1 mM EGTA, 1 mM DTT, 20 µM taxol, with 1 mM Mg-ATP); (2) 0.5 mg ml$^{-1}$ neutravidin for 2 min, followed by two washes with buffer B; (3) biotinylated motor protein (100 nM) for 2 min, followed by two washes with buffer B supplemented with 1 mg ml$^{-1}$ casein; (4) appropriately diluted Alexa Fluor 488 microtubules in assay solution (buffer B supplemented with 1 mg ml$^{-1}$ casein, 1 mM Mg-ATP, 71 mM β-mercaptoethanol, 20 mM glucose, 300 µg ml$^{-1}$ glucose oxidase, 60 µg ml$^{-1}$ catalase).

For single-molecule assays with Kif3AB/C and Kap3, steps 1 and 2 were carried out as described above. Then, chambers were incubated with: (3) appropriately diluted Alexa Fluor 488, biotinylated microtubules for 2 min, followed by two washes with buffer B supplemented with 1 mg ml$^{-1}$ casein; and (4) 0.15–40 nM of labeled motor protein (Alexa Fluor 647 or TMR) in assay solution (buffer B with supplements as above, and 1 mM Mg-ATP or AMPNP as indicated). For Kap3 colocalization assays, the Kap3 concentration was 1 nM. Motility assays with APC$^{ARM/ARM-Mut}$ were carried out in buffer C (80 mM PIPES pH 6.9, 2 mM MgCl$_2$, 1 mM EGTA, 1 mM DTT, 20 µM taxol, 1 mM Mg-ATP), with the same supplements as for buffer B. Kif3AB and Kap3 were incubated on ice for 10 min before being added, and, if included, APC$^{ARM/ARM-Mut}$ was incubated for a further 5 min (final concentrations: 40 nM Kif3AB–Kap3, 1.25 µM APC$^{ARM/ARM-Mut}$). Because Kif3AB has the propensity to aggregate into motile clusters (distinguishable from single molecules by their bright fluorescence), even if autoinhibited at the single-molecule level, great care must be taken to remove protein aggregates before TIRF microscopy (ultracentrifugation in a TLA 100 rotor at 386,400$g$ for 10 min or size-exclusion chromatography).

### TIRF microscopy

Fluorescently-labeled molecules were visualized on an Eclipse Ti-E inverted microscope with a CFI Apo TIRF 1.49-N.A. oil objective, Perfect Focus System, H-TIRF module, LU-N4 laser unit (Nikon) and a quad band filter set (Chroma). Images were recorded with 100 ms exposures on an iXon DU888 Ultra EMCCD camera (Andor), controlled with NIS-Elements AR Software (Nikon). The microscope was kept in a temperature-controlled environmental chamber (Okolab) operating at 25 °C for in vitro assays and 37 °C for live-cell imaging[69]. Files were imported into Fiji[70] (ImageJ, NIH) for analysis.

Velocities and run lengths were calculated from kymographs generated in Fiji. Microtubules that were not fully enclosed in the field of view or that overlapped with another microtubule were excluded. Average run lengths were determined by fitting a one-phase exponential decay to the cumulative frequency distribution of run lengths. For single-molecule velocity measurements, events longer than 4 consecutive pixels were analyzed. For measurement of Kap3 colocalization, a circular region of interest was drawn around each Kif3A spot in the 640-nm channel and assessed for signal in the Kap3 561-nm channel. Graphing, curve fitting and statistical analysis were performed in Prism9 and Prism10 (GraphPad).

## Electron microscopy

Reconstituted Kif3AB–Kap3 and Kif3AB–Kap–APC$^{ARM}$ samples were diluted to 30–100 nM in buffer A without glycerol, and 4 µl of specimen was added onto a glow-discharged continuous carbon grid (Electron Microscopy Sciences). Grids were then stained in three sequential drops of 2% uranyl acetate, blotted and air dried. Kif3AB–Kap3 data were collected on a Tecnai T12 microscope (Thermo Fisher Scientific) operating at 120 kV and a 4k × 4k CCD camera (Gatan US4000) at a nominal magnification of ×52,000, resulting in a sampling of 2.09 Å per pixel. Kif3AB–Kap–APC$^{ARM}$ data were collected on a JEM-1400Flash microscope (JEOL) operating at 120 kV and a 4k × 4k CMOS Rio camera (Gatan) at nominal magnification of ×40,000, resulting in a sampling of 1.4 Å per pixel. Subsequent image processing of Kif3AB–Kap3 data was carried out in Cryosparc[71] unless stated otherwise. Particles were picked from micrographs using blob picker, followed by template picker once initial 2D classes were generated. Particles were extracted into 256-pixel boxes and subjected to multiple rounds of 2D classification, from which a subset of well-resolved classes encompassing 3,032 individual particles was obtained. These classes were then compared with projections of the Kif3AB–Kap3 AlphaFold3 model. For this, the atomic model was converted to a density map in UCSF Chimera[72] and low-pass filtered to 25 Å using EMAN2 (ref. 73). This volume was used to generate 3,032 projections in Cryosparc, using the simulate data job, and these were then classified into 50 2D classes. These classes were aligned to each data class average and scored by cross-correlation to identify the best matching projection in SPIDER[74].

## AlphaFold

AlphaFold models were generated for full-length sequences using Alphafold3 (ref. 75) and the ColabFold[76] implementation of AlphaFold2 (ref. 77) and AlphaFold Multimer[78]. Visualization and structural alignment were carried out in UCSF Chimera and Chimera X[79].

## Construct generation and cell biology experiments

For cell biology experiments, mouse IMCD-3-FlpIn cells (gift from P. K. Jackson, Stanford)[80] were cultured in DMEM/F12 (Gibco) supplemented with 10% FBS and 100 U ml$^{-1}$ penicillin–streptomycin. Cells were incubated in serum-free medium for 24 h to induce ciliogenesis.

For CRISPR–Cas9 genome editing in IMCD-3-FlpIn cells to generate a Kif3B KO, guide RNA (5′-AAGCTCAGAATCAGTCCGGG-3′) targeting exon 2 of Kif3B was designed in Benchling and cloned into the pX330 Cas9 plasmid (gift from F. Zhang; Addgene plasmid no. 42230)[81]. IMCD-3 cells were transfected with 2.4 µg DNA (1.2 µg Kif3B guide expressing plasmid and 1.2 µg mScarlet vector) using the Lipofectamine 2000 transfection reagent (Thermo Fisher Scientific) in Opti-MEM reduced serum medium (Thermo Fisher Scientific)[69]. A population of mScarlet-positive cells was collected through fluorescence-activated cell sorting. These cells were cultured for 2 weeks before single cell sorting and subsequent cell culture expansion.

To validate knockout of Kif3B, genomic DNA was isolated (Lucigen, QE0905T) and targeted sequences were PCR amplified, cloned into pJET1.2/blunt vector (Thermo Fisher Scientific, K1232) and subsequently sequenced.

To stably express mScarlet-tagged Kif3B constructs in Kif3B-KO IMCD-3 cells, the Super PiggyBac transposon vector system (System Biosciences) was used. Cells in six-well plates were cotransfected with PiggyBac plasmid containing an mScarlet-tagged gene of interest (Kif3B$^{WT}$, Kif3B$^{ΔC156}$ and Kif3B$^{KKK}$) with a geneticin resistance marker (0.5 µg) and Super PiggyBac transposase expression vector (0.2 µg) using Lipofectamine 2000. Clones were selected using geneticin resistance (500 µg ml$^{-1}$) 2 days after transfection, cultured until confluent and screened by live-cell TIRF microscopy and immunoblotting to confirm the expression of mScarlet-labeled proteins. For continued culture, growth medium contained 500 µg ml$^{-1}$ geneticin.

For Kap3 localization experiments, an mNeonGreen-tagged Kap3 construct was stably expressed using the PiggyBac transposon vector

(blasticidin resistance) in either the Kif3B$^{WT}$ or Kif3B$^{KKK}$ IMCD-3 cell lines, generated as described above. Clones were selected using blasticidin (10 µg ml$^{-1}$) 2 days after transfection, cultured until confluent and screened by live-cell TIRF microscopy to confirm expression of mNeonGreen–Kap3. For continued culture, growth medium contained 500 µg ml$^{-1}$ geneticin and 10 µg ml$^{-1}$ blasticidin.

## Immunoblotting

Cells were lysed in RIPA buffer containing 150 mM sodium chloride, 1.0% Triton X-100, 0.5% sodium deoxycholate, 0.1% SDS and 50 mM Tris pH 8.0. Samples were separated by SDS–PAGE, followed by transfer to nitrocellulose membranes. Membranes were blocked overnight in 3% (wt/vol) milk:TBS-T (20 mM Tris-base, 150 mM NaCl, 0.02% Tween 20) and incubated for 4 h in anti-Flag M2 mouse antibody (1:1,000; Sigma-Aldrich, F1804) or anti-GAPDH (1:10,000; Cell Signaling Technology, 2118), as previously described[69,82]. After incubation, membranes were washed for 4 × 5 min with TBS-T, before incubation with goat anti-mouse-IgG StarBright Blue 700 (1:1,000; BioRad) or goat anti-rabbit-IgG (H+L) Alexa Fluor 647 (1:1,000; Invitrogen) secondary antibodies in 2% (wt/vol) milk:TBS-T for 1 h at room temperature. Blots were then washed with TBS-T for 4 × 5 min before imaging.

## Immunofluorescence

IMCD-3 cells grown on 0.17-mm-thick (no. 1.5) cover glasses (VWR) were washed with PBS, followed by two washes in cytoskeletal buffer (100 mM NaCl, 300 mM sucrose, 3 mM MgCl$_2$, 10 mM PIPES pH 6.9) and fixed in 4% paraformaldehyde prepared in cytoskeletal buffer with 0.5% Triton and 5 mM EGTA[69]. Cells were blocked in 3% BSA and 2% FBS in PBS for 1 h at room temperature and incubated overnight with the respective antibodies. After incubation with primary antibodies, cells were washed and incubated with the corresponding secondary antibody in a blocking solution for 1 h. Coverslips were mounted onto glass slides using mounting medium (GeneTex) and imaged by TIRF microscopy. The following antibodies were used at the indicated dilutions: anti-acetylated tubulin (Sigma-Aldrich T6793; 1:2,000); anti-gamma-tubulin (Sigma-Aldrich T6557; 1:500). Alexa-Fluor-labeled secondary antibodies (Thermo Fisher) were used at a 1:500 dilution.

Immunofluorescence and live-cell images were analyzed in Fiji. Cilia length was measured using the 'segmented line' tool. Time-averaged fluorescence distributions of mScarlet Kif3B and mNeonGreen–Kap3 were generated by Z-projecting time-lapse images, subtracting background and tracing cilia using the segmented line tool. The 'plot profile' tool was used to obtain fluorescence intensities profiles[69,82].

## Statistics and reproducibility

Exact P values are as follows. Figure 2c: one-way ANOVA followed by Tukey's multiple comparison test. Velocity: Kif3A$^{Δ89}$B versus Kif3A$^{Δ103}$B $P = 0.9980$; Kif3A$^{Δ89}$B versus Kif3A$^{Δ103}$B$^{Δ39}$ $P = 0.6827$; Kif3A$^{Δ89}$B versus Kif3A$^{Δ103}$B$^{Δ90}$ $P > 0.9999$; Kif3A$^{Δ89}$B versus Kif3A$^{Δ103}$B$^{Δ114}$ $P > 0.9999$; Kif3A$^{Δ89}$B versus Kif3A$^{Δ103}$B$^{Δ156}$ $P = 0.7180$; Kif3A$^{Δ103}$B versus Kif3A$^{Δ103}$B$^{Δ39}$ 0.4495; Kif3A$^{Δ103}$B versus Kif3A$^{Δ103}$B$^{Δ90}$ $P > 0.9999$; Kif3A$^{Δ103}$B versus Kif3A$^{Δ103}$B$^{Δ114}$ $P = 0.9907$; Kif3A$^{Δ103}$B versus Kif3A$^{Δ103}$B$^{Δ156}$ $P = 0.90907$; Kif3A$^{Δ103}$B$^{Δ39}$ versus Kif3A$^{Δ103}$B$^{Δ90}$ $P = 0.5664$; Kif3A$^{Δ103}$B$^{Δ39}$ versus Kif3A$^{Δ103}$B$^{Δ114}$ $P = 0.7717$; Kif3A$^{Δ103}$B$^{Δ39}$ versus Kif3A$^{Δ103}$B$^{Δ156}$ $P = 0.1074$; Kif3A$^{Δ103}$B$^{Δ90}$ versus Kif3A$^{Δ103}$B$^{Δ114}$ $P = 0.9989$; Kif3A$^{Δ103}$B$^{Δ90}$ versus Kif3A$^{Δ103}$B$^{Δ156}$ $P = 0.8229$; Kif3A$^{Δ103}$B$^{Δ114}$ versus Kif3A$^{Δ103}$B$^{Δ156}$ $P = 0.6256$. Run-length: Kif3A$^{Δ89}$B versus Kif3A$^{Δ103}$B $P > 0.9999$; Kif3A$^{Δ89}$B versus Kif3A$^{Δ103}$B$^{Δ39}$ $P > 0.9999$; Kif3A$^{Δ89}$B versus Kif3A$^{Δ103}$B$^{Δ90}$ $P > 0.8358$; Kif3A$^{Δ89}$B versus Kif3A$^{Δ103}$B$^{Δ114}$ $P = 0.8851$; Kif3A$^{Δ89}$B versus Kif3A$^{Δ103}$B$^{Δ156}$ $P = 0.9281$; Kif3A$^{Δ103}$B versus Kif3A$^{Δ103}$B$^{Δ39}$ $P > 0.9999$; Kif3A$^{Δ103}$B versus Kif3A$^{Δ103}$B$^{Δ90}$ $P = 0.8736$; Kif3A$^{Δ103}$B versus Kif3A$^{Δ103}$B$^{Δ114}$ $P = 0.91607$; Kif3A$^{Δ103}$B versus Kif3A$^{Δ103}$B$^{Δ156}$ $p = 0.8996$; Kif3A$^{Δ103}$B$^{Δ39}$ versus Kif3A$^{Δ103}$B$^{Δ90}$ $p = 0.7832$; Kif3A$^{Δ103}$B$^{Δ39}$ versus Kif3A$^{Δ103}$B$^{Δ114}$ $p = 0.8398$; Kif3A$^{Δ103}$B$^{Δ39}$ versus Kif3A$^{Δ103}$B$^{Δ156}$ $P = 0.9556$; Kif3A$^{Δ103}$B$^{Δ90}$ versus Kif3A$^{Δ103}$B$^{Δ114}$ $P > 0.9999$; Kif3A$^{Δ103}$B$^{Δ90}$ versus Kif3A$^{Δ103}$B$^{Δ156}$

$P = 0.3370$; Kif3A$^{\Delta103}$B$^{\Delta114}$ versus Kif3A$^{\Delta103}$B$^{\Delta156}$ $P = 0.3918$. Landing rate: Kif3A$^{\Delta89}$B versus Kif3A$^{\Delta103}$B $P > 0.9999$; Kif3A$^{\Delta89}$B versus Kif3A$^{\Delta103}$B$^{\Delta39}$ $P > 0.9999$; Kif3A$^{\Delta89}$B versus Kif3A$^{\Delta103}$B$^{\Delta90}$ $P = 0.0002$; Kif3A$^{\Delta89}$B versus Kif3A$^{\Delta103}$B$^{\Delta114}$ $P = 0.0001$; Kif3A$^{\Delta89}$B versus Kif3A$^{\Delta103}$B$^{\Delta156}$ $P = 0.0014$; Kif3A$^{\Delta103}$B versus Kif3A$^{\Delta103}$B$^{\Delta39}$ $P > 0.9999$; Kif3A$^{\Delta103}$B versus Kif3A$^{\Delta103}$B$^{\Delta90}$ $P = 0.0002$; Kif3A$^{\Delta103}$B versus Kif3A$^{\Delta103}$B$^{\Delta114}$ $P = 0.0001$; Kif3A$^{\Delta103}$B versus Kif3A$^{\Delta103}$B$^{\Delta156}$ $P = 0.0017$; Kif3A$^{\Delta103}$B$^{\Delta39}$ versus Kif3A$^{\Delta103}$B$^{\Delta90}$ $P = 0.0002$; Kif3A$^{\Delta103}$B$^{\Delta39}$ versus Kif3A$^{\Delta103}$B$^{\Delta114}$ $P = 0.0001$; Kif3A$^{\Delta103}$B$^{\Delta39}$ versus Kif3A$^{\Delta103}$B$^{\Delta156}$ $P = 0.0015$; Kif3A$^{\Delta103}$B$^{\Delta90}$ versus Kif3A$^{\Delta103}$B$^{\Delta114}$ $P = 0.9993$; Kif3A$^{\Delta103}$B$^{\Delta90}$ versus Kif3A$^{\Delta103}$B$^{\Delta156}$ $P = 0.7208$; Kif3A$^{\Delta103}$B$^{\Delta114}$ versus Kif3A$^{\Delta103}$B$^{\Delta156}$ $P = 0.5319$. Figure 3a: one-way ANOVA followed by Tukey's multiple comparison test. Control versus Kif3B KO $P < 0.0001$; control versus Kif3B KO + Kif3B$^{WT}$ $P = 0.5039$; control versus Kif3B KO + Kif3B$^{\Delta C156}$ $P < 0.0001$; Control versus Kif3B KO + Kif3B$^{KKK}$ $P = 0.4914$; Kif3B KO versus Kif3B KO + Kif3B$^{WT}$ $P < 0.0001$; Kif3B KO versus Kif3B KO + Kif3B$^{\Delta C156}$ $P = 0.9997$; Kif3B KO versus Kif3B KO + Kif3B$^{KKK}$ $P < 0.0001$; Kif3B KO + Kif3B$^{WT}$ versus Kif3B KO + Kif3B$^{\Delta C156}$ $P < 0.0001$; Kif3B KO + Kif3B$^{WT}$ versus Kif3B KO + Kif3B$^{KKK}$ $P > 0.9999$; Kif3B KO + Kif3B$^{\Delta C156}$ versus Kif3B KO + Kif3B$^{KKK}$ $P < 0.0001$. Figure 4g: one-way ANOVA followed by Tukey's multiple comparison test. WT versus $\Delta$Int3 $P = 0.0915$; WT versus $\Delta$Int2+3 $P = 0.0071$; WT versus $\Delta$Int1 $P < 0.0001$; WT versus $\Delta$Int1–3 $P < 0.0001$; $\Delta$Int3 versus $\Delta$Int2+3 $P = 0.4892$; $\Delta$Int3 versus $\Delta$Int1 $P = 0.0034$; $\Delta$Int3 versus $\Delta$Int1–3 $P < 0.0001$; $\Delta$Int2+3 versus $\Delta$Int1 $P = 0.0422$; $\Delta$Int2+3 versus $\Delta$Int1–3 $P = 0.0003$; $\Delta$Int1 versus $\Delta$Int1–3 $P = 0.0342$.

## Materials availability

All unique biological materials used are available from the authors or from standard commercial sources.

## Reporting summary

Further information on research design is available in the Nature Portfolio Reporting Summary linked to this article.

## Data availability

Unprocessed gels and immunoblots and numerical source data are available in Source Data, provided with this paper. AlphaFold models are available at https://doi.org/10.5281/zenodo.15707065. All other data supporting the conclusions of this study are available from the corresponding author upon reasonable request. Source data are provided with this paper.

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

## Acknowledgements

We thank A. Carver, F. Gonçalves-Santos and W. Allen (Sir William Dunn School of Pathology) for comments on the manuscript; C. Moores (Birkbeck) for discussions; N. Lukoyanova, S. Chen (Birkbeck), C. Melia and R. Dhaliwal (Sir William Dunn School of Pathology) for EM support; D. Houldershaw, Y. Goudetsidis, R. Westlake (Birkbeck) and E. Lowe (COSMIC) for computational support; and C. Studniarek (Sir William Dunn School of Pathology) for assistance with Western blotting. This work is funded by grants from the Wellcome Trust (214998/Z/18/Z and 217186/Z/19/Z), UKRI Biotechnology and Biological Sciences Research Council (BB/P008348/1) and UKRI Medical Research Council (MR/Z504750/1).

## Author contributions

Methodology, S.W., K.T., A.G.M., S.D.N. and A.J.R.; Investigation, S.W., K.T., A.G.M., S.D.N. and A.J.R.; Writing – Original Draft, K.T., A.G.M. and A.J.R.; Writing – Review & Editing, S.W., K.T., A.G.M., S.D.N. and A.J.R. Funding Acquisition, K.T. and A.J.R.

## Competing interests

The authors declare no competing interests.

## Additional information

**Extended data** is available for this paper at https://doi.org/10.1038/s41594-025-01630-5.

**Correspondence and requests for materials** should be addressed to Anthony J. Roberts.

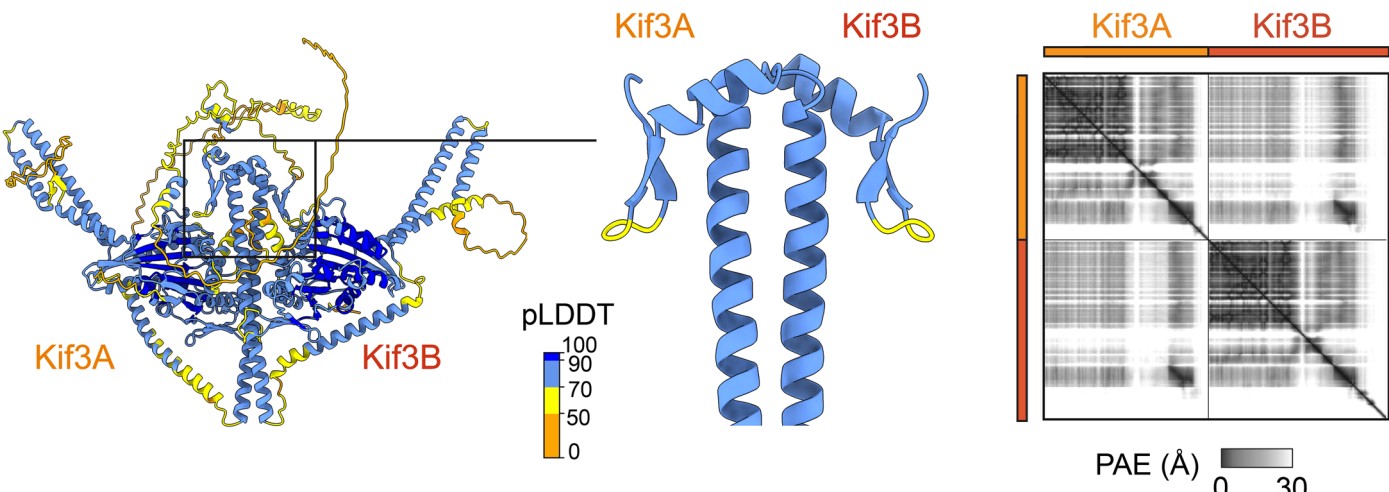

**Extended Data Fig. 1 | AlphaFold3 analysis of Kif3AB.** Left, Kif3AB AlphaFold3 model showing β-hairpin motif, colored by pLDDT (predicted local distance difference test) according to the key. Right, predicted aligned error (PAE) plot of Kif3AB, full-length proteins.

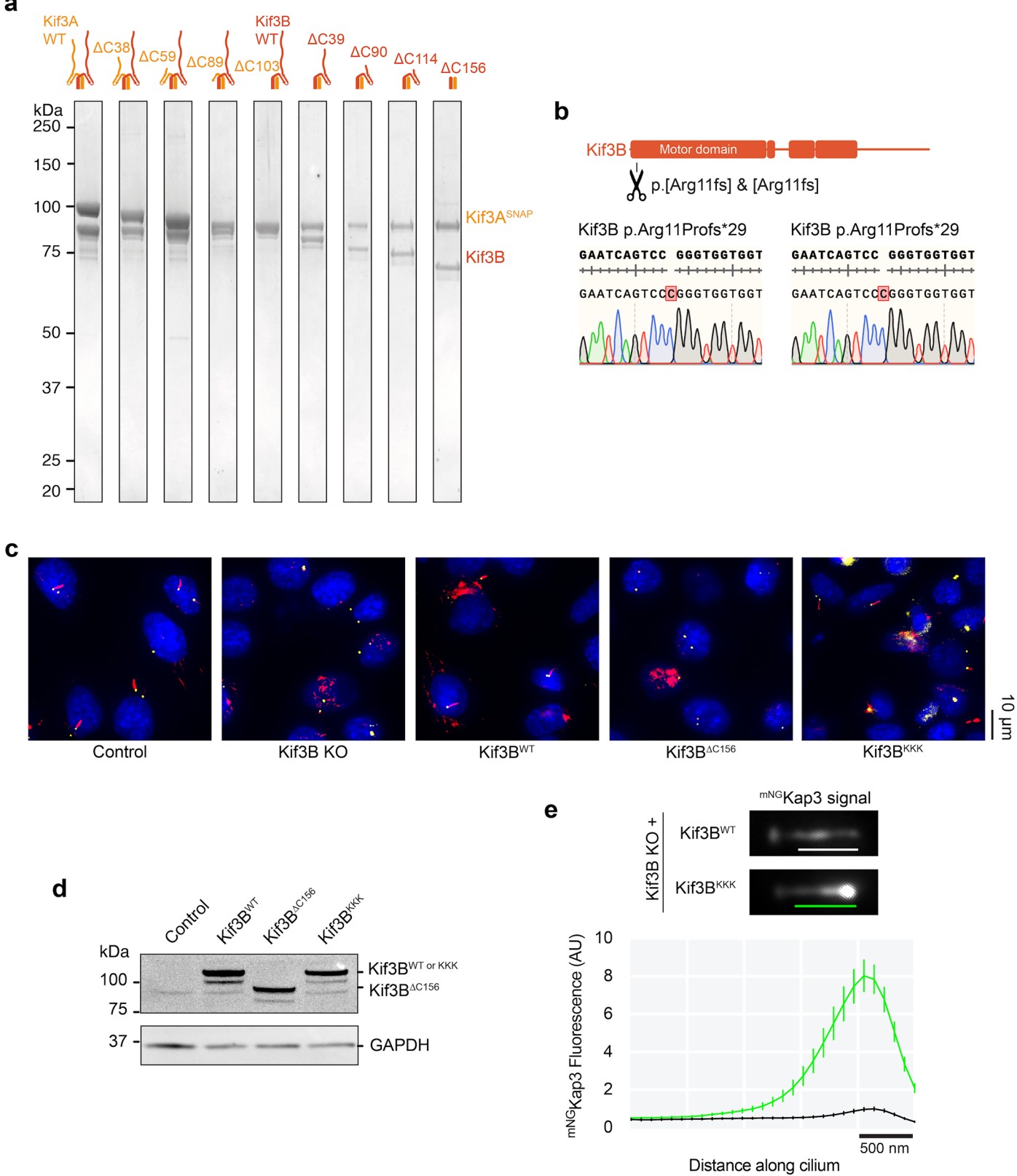

Extended Data Fig. 2 | See next page for caption.

**Extended Data Fig. 2 | Purification of Kif3AB constructs, CRISPR knockout of Kif3B, and Kap3 imaging in cilia. a**, SDS-PAGE of purified Kif3AB constructs (post size-exclusion chromatography) with indicated C-terminal truncations. **b**, Genotype for homozygous Kif3B KO cell line, with insertions highlighted by alignment with the reference sequence. Clones were extensively Sanger sequenced to determine genotype (representative traces shown). **c**, Immunofluorescence images of cilia in Kif3B KO cell lines and cell lines stably expressing Kif3B^WT, Kif3B^ΔC156 and Kif3B^KKK. Cells were stained for gamma tubulin (yellow), acetylated tubulin (red) and DAPI (blue). **d**, Western blot showing expression of FLAG-tagged Kif3B^WT, Kif3B^ΔC156 or Kif3B^KKK in Kif3B KO cells, detected using anti-FLAG. GAPDH used as loading control. **e**. Top, representative images of mNeonGreen (mNG) tagged Kap3 in background of unlabelled Kif3B^WT and Kif3B^KKK constructs expressed in Kif3B KO cells. Bottom, plot of average mNG-Kap3 fluorescent signal from line scans along the cilium length (bars in top panel indicate distance analyzed). Kif3B^WT background, black. Kif3B^KKK background, green. Values are normalized relative to Kif3B^WT peak value. Traces show mean intensity ± s.e.m.; n = 39 (Kif3B^WT), 31 (Kif3B^KKK) cilia measured from three technical replicates.

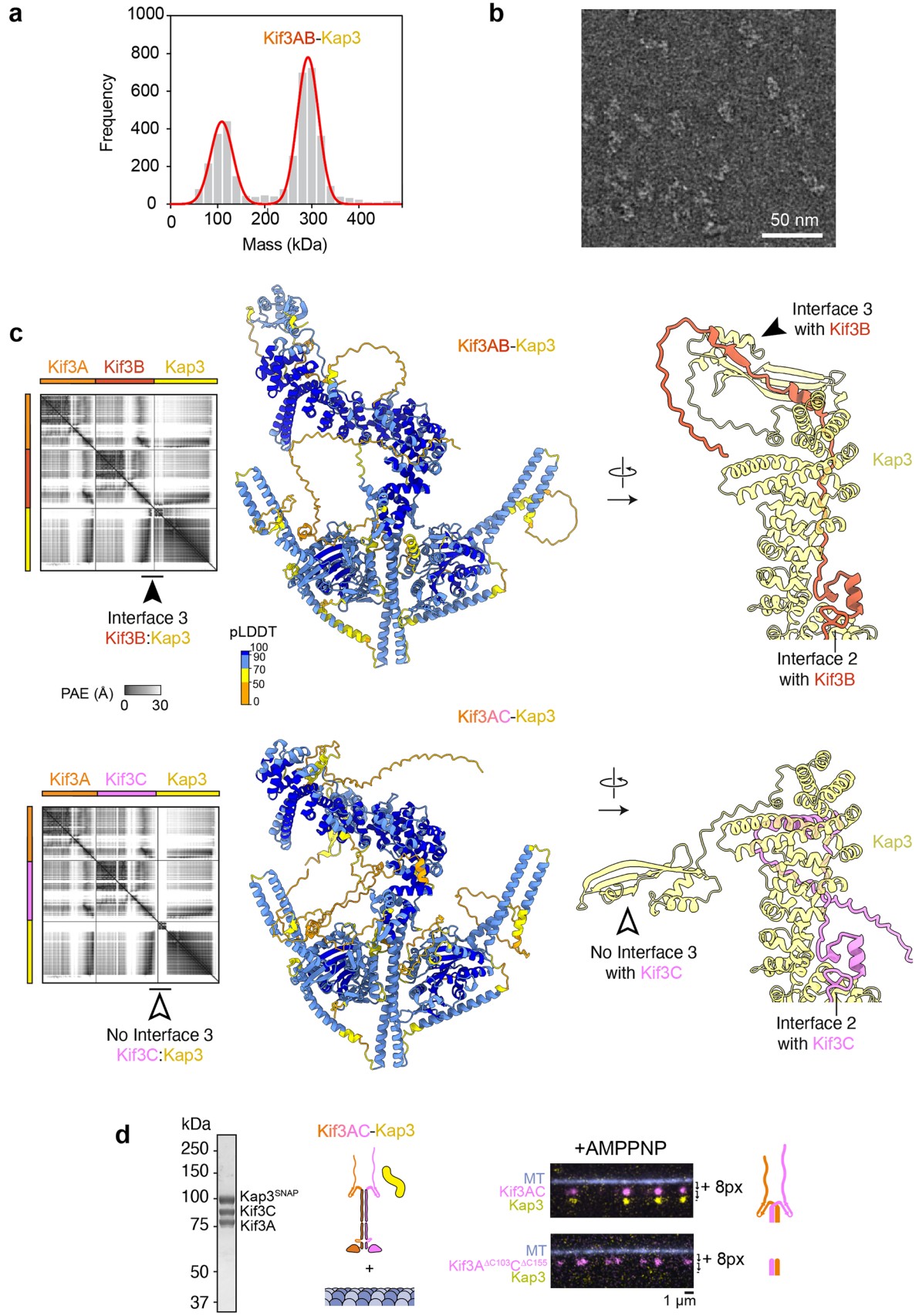

**Extended Data Fig. 3 | See next page for caption.**

**Extended Data Fig. 3 | Characterization of the Kif3AB-Kap3 and Kif3AC-Kap3 heterotrimer. a**, Mass photometry of purified Kif3AB-Kap3. Main peak (291 ± 23 kDa, mean ± s.d.) is consistent with the theoretical mass of heterotrimer (278 kDa). **b**, Electron micrograph of negatively stained Kif3AB-Kap3. **c**, Left, AlphaFold3 models of Kif3AB-Kap3 and Kif3AC-Kap3 colored by pLDDT according to the key. PAE plots alongside. Interface 3 highlighted with black arrowhead for Kif3B and lack of equivalent interface with white arrowhead for Kif3C. Right, Close-up views of Kap3 for each model. Note interface 2 with Kap3 common to Kif3B and Kif3C and lack of interface 3 with Kif3C (black versus white arrowheads). **d**, Left, SDS-PAGE of purified Kif3AC-Kap3 heterotrimer following size-exclusion chromatography, peak fraction shown. Right, Kap3 co-localization assay with full-length Kif3AC (top) and with Kif3AC lacking C-terminal regions (Kif3A$^{\Delta C103}$C$^{\Delta C155}$; bottom). Composite TIRF images are shown of 488 nm channel (MT), 640 nm channel (Alexa-Fluor-647-labeled Kif3A) offset in y-axis by 8 pixels, and 561 nm channel (TMR-labeled Kap3) offset in y-axis by 16-pixels.

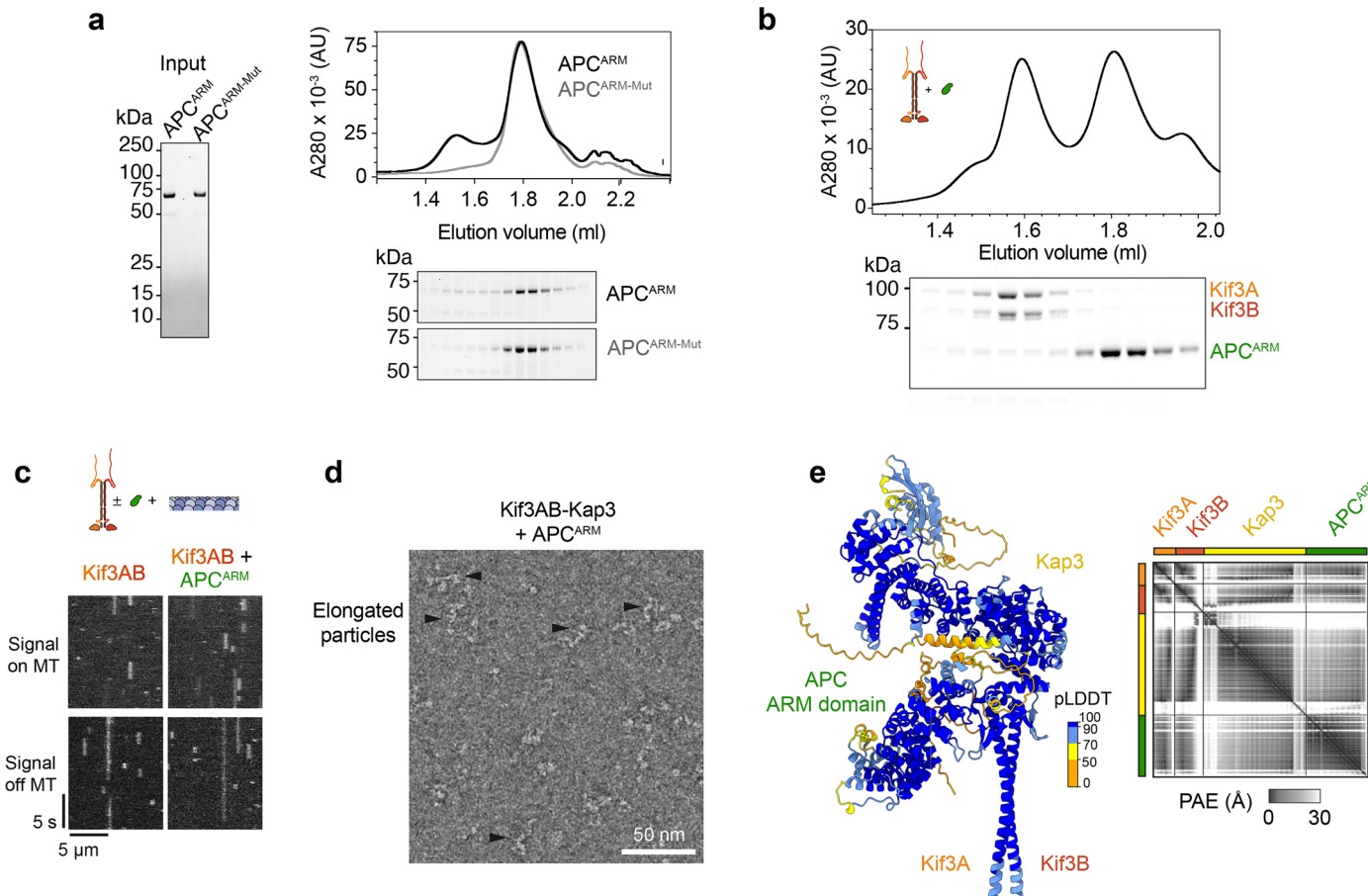

**Extended Data Fig. 4 | Characterization of APC^ARM and interaction with Kif3AB-Kap3. a**. Size-exclusion chromatogram of purified APC^ARM and APC^ARM-Mut proteins. SDS-PAGE of input shown left and of fractions beneath. **b**, Size-exclusion chromatogram of binding reaction between Kif3AB and APC^ARM with SDS-PAGE of fractions beneath. Note lack of co-elution of APC^ARM with Kif3AB compared to when Kap3 is present (Fig. 6a). **c**, Examples kymographs of Kif3AB (Alexa-Fluor-647-labeled Kif3A) in presence of microtubules and APC^ARM (unlabeled). As there is non-specific binding of Kif3AB to the coverslip surface in this experiment, kymographs are shown for regions on and off the microtubule. APC^ARM does not activate the motility of Kif3AB in the absence of Kap3. **d**, Electron micrograph of negatively stained Kif3AB-Kap3 in presence of APC^ARM. Particles with an elongated appearance indicated with arrowheads. **e**. AlphaFold3 model of Kif3AB-Kap3-APC^ARM colored by pLDDT according to the key. Right, PAE plot.

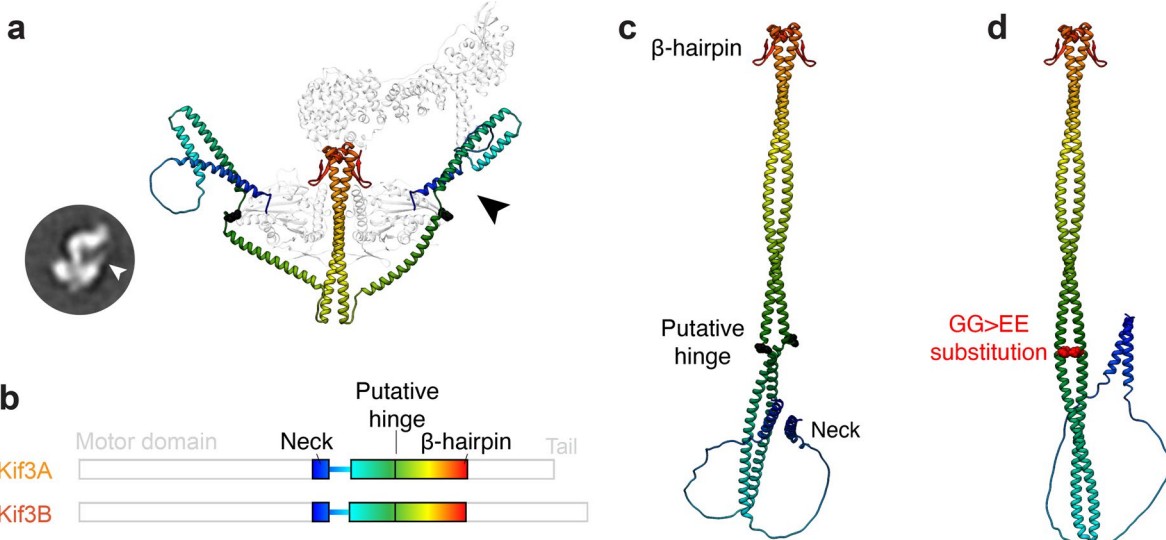

**Extended Data Fig. 5 | Analysis of the coiled-coil regions in Kif3AB. a**, Kif3AB-Kap3 AlphaFold3 model with predicted coiled coil regions colored in rainbow from N-terminus (blue) to C-terminus (red). Note the predicted separation of the strands of the N-terminal coiled coil in the autoinhibited state, resulting in a fine structure protruding from the motor domain (black arrowhead). This model is consistent with EM class averages showing equivalent fine protruding structure (white arrowhead, inset). **b**, Sequence diagrams of Kif3A and Kif3B, with predicted coiled coil regions colored in rainbow as in panel a. The location of the putative hinge in the coiled coil (Kif3A- G485 G486; Kif3B-G477 G478) is indicated. **c**, AlphaFold3 model of the Kif3AB coiled coil region lacking the motor domains (thereby preventing the formation of the autoinhibited conformation). This model shows an extended coiled coil, with a hinge at the predicted location, and may represent conformation of the coiled coil in active Kif3AB. Note that transition between the autoinhibited conformation in panel a and the extended conformation in panel c is not well described by simple folding at putative hinge. **d**, Mutation of the α-helix-breaking glycine residues at the putative hinge (Kif3A-G485 G486; Kif3B-G477 G478) to a residue with higher α-helical propensity (glutamic acid; E) is known to relieve autoinhibition in kinesin-2 (Brunnbauer et al. *PNAS* 107, 10460–10465, 2010). An AlphaFold3 model of the Kif3AB coiled coil region including the equivalent substitutions (Kif3A-G485E G486E; Kif3B-G477E G478E – GG > EE) shows a continuous coiled coil rather than a hinge, which could prevent access of the motor domains to the β-hairpin motif, consistent with the observed activation.

**Extended Data Table 1 | Motility parameters for Kif3AB constructs**

| Construct | Velocity (nm s⁻¹) | Landing rate (µm⁻¹ nM⁻¹ min⁻¹) | Run length (µm) |
|---|---|---|---|
| Kif3A-Kif3B | – | – | – |
| Kif3A$^{\Delta C38}$-Kif3B | – | – | – |
| Kif3A$^{\Delta C59}$-Kif3B | – | – | – |
| Kif3A$^{\Delta C89}$-Kif3B | 945 ± 30 | 0.55 ± 0.04 | 0.27 ± 0.01 |
| Kif3A$^{\Delta C103}$-Kif3B | 933 ± 23 | 0.60 ± 0.07 | 0.27 ± 0.02 |
| Kif3A$^{\Delta C103}$-Kif3B$^{\Delta C39}$ | 990 ± 4 | 0.56 ± 0.06 | 0.28 ± 0.03 |
| Kif3A$^{\Delta C103}$-Kif3B$^{\Delta C90}$ | 939 ± 4 | 3.30 ± 0.20 | 0.20 ± 0.02 |
| Kif3A$^{\Delta C103}$-Kif3B$^{\Delta C114}$ | 950 ± 15 | 3.40 ± 0.26 | 0.21 ± 0.01 |
| Kif3A$^{\Delta C103}$-Kif3B$^{\Delta C156}$ | 903 ± 30 | 2.70 ± 0.18 | 0.33 ± 0.09 |
| Kif3A$^{KK}$-Kif3B | 730 ± 71 | – | 0.55 ± 0.04 |
| Kif3A-Kif3B$^{KKK}$ | 740 ± 75 | – | 0.64 ± 0.04 |
| Kif3A$^{KK}$-Kif3B$^{KKK}$ | 693 ± 82 | – | 0.60 ± 0.06 |
| Kif3A$^{Gly-mut}$-Kif3B | 964 ± 14 | – | 0.54 ± 0.01 |
| Kif3A$^{CC-mut}$Kif3B$^{CC-mut}$ | 929 ± 13 | – | 0.79 ± 0.03 |
| Kif3A-Kif3B-Kap3-APC | 801 ± 21 | 0.09 ± 0.01 | 0.47 ± 0.02 |

Values show mean ± SEM from 3 technical replicates. More than 100 motile events were analyzed per replicate (exact *n* values given in main figure legends). No single-molecule motile events were observed for Kif3A-Kif3B, Kif3A$^{\Delta C38}$-Kif3B, or Kif3A$^{\Delta C59}$-Kif3B, suggestive of strong autoinhibition. Landing rates not given for Kif3A$^{KK}$-Kif3B, Kif3A-Kif3B$^{KKK}$, Kif3A$^{KK}$-Kif3B$^{KKK}$, Kif3A$^{Gly-mut}$-Kif3B, and Kif3A$^{CC-mut}$Kif3B$^{CC-mut}$ as precise protein concentrations not measurable (low protein yield). Landing rate for Kif3A$^{KK}$-Kif3B, Kif3A-Kif3B$^{KKK}$, Kif3A$^{KK}$-Kif3B$^{KKK}$, and Kif3A$^{Gly-mut}$-Kif3B broadly comparable to Kif3A-Kif3B-Kap3-APC.

# Reporting Summary

## Statistics

For all statistical analyses, confirm that the following items are present in the figure legend, table legend, main text, or Methods section.

| n/a | Confirmed | |
|---|---|---|
| ☐ | ☒ | The exact sample size (*n*) for each experimental group/condition, given as a discrete number and unit of measurement |
| ☐ | ☒ | A statement on whether measurements were taken from distinct samples or whether the same sample was measured repeatedly |
| ☐ | ☒ | The statistical test(s) used AND whether they are one- or two-sided *Only common tests should be described solely by name; describe more complex techniques in the Methods section.* |
| ☒ | ☐ | A description of all covariates tested |
| ☒ | ☐ | A description of any assumptions or corrections, such as tests of normality and adjustment for multiple comparisons |
| ☐ | ☒ | A full description of the statistical parameters including central tendency (e.g. means) or other basic estimates (e.g. regression coefficient) AND variation (e.g. standard deviation) or associated estimates of uncertainty (e.g. confidence intervals) |
| ☐ | ☒ | For null hypothesis testing, the test statistic (e.g. *F*, *t*, *r*) with confidence intervals, effect sizes, degrees of freedom and *P* value noted *Give P values as exact values whenever suitable.* |
| ☒ | ☐ | For Bayesian analysis, information on the choice of priors and Markov chain Monte Carlo settings |
| ☒ | ☐ | For hierarchical and complex designs, identification of the appropriate level for tests and full reporting of outcomes |
| ☒ | ☐ | Estimates of effect sizes (e.g. Cohen's *d*, Pearson's *r*), indicating how they were calculated |

*Our web collection on statistics for biologists contains articles on many of the points above.*

## Software and code

Policy information about availability of computer code

| Data collection | Total internal reflelection fluorescence microscopy data were collected with NIS-Elements AR software v4.60.00 (Nikon). Mass photometry data were collected with AcquireMP v2 (Refeyn) and processed with DiscoverMP v2 (Refeyn) software. Electron microscopy data were collected using TUI software 2017 (ThermoFisher Scientific) and DigitalMicrograph software 1.85.1535 (Gatan). |
|---|---|
| Data analysis | SPIDER v22.10, EMAN2 v2.91, Cryosparc v4.5.1, UCSF Chimera v1.16, UCSF ChimeraX v1.8, Fiji v2.9.0, GraphPad Prism v9 and v10. |

For manuscripts utilizing custom algorithms or software that are central to the research but not yet described in published literature, software must be made available to editors and reviewers. We strongly encourage code deposition in a community repository (e.g. GitHub). See the Nature Portfolio guidelines for submitting code & software for further information.

## Data

Policy information about availability of data

All manuscripts must include a data availability statement. This statement should provide the following information, where applicable:
- Accession codes, unique identifiers, or web links for publicly available datasets
- A description of any restrictions on data availability
- For clinical datasets or third party data, please ensure that the statement adheres to our policy

All data supporting the conclusions of this manuscript are available from the corresponding author upon reasonable request.

# Research involving human participants, their data, or biological material

Policy information about studies with <u>human participants or human data</u>. See also policy information about <u>sex, gender (identity/presentation), and sexual orientation</u> and <u>race, ethnicity and racism</u>.

| | |
|---|---|
| Reporting on sex and gender | NA |
| Reporting on race, ethnicity, or other socially relevant groupings | NA |
| Population characteristics | NA |
| Recruitment | NA |
| Ethics oversight | NA |

Note that full information on the approval of the study protocol must also be provided in the manuscript.

# Field-specific reporting

Please select the one below that is the best fit for your research. If you are not sure, read the appropriate sections before making your selection.

☒ Life sciences  ☐ Behavioural & social sciences  ☐ Ecological, evolutionary & environmental sciences

For a reference copy of the document with all sections, see <u>nature.com/documents/nr-reporting-summary-flat.pdf</u>

# Life sciences study design

All studies must disclose on these points even when the disclosure is negative.

| | |
|---|---|
| Sample size | No statistical methods were used to predetermine sample size. The number of motile events analyzed per construct was based on previous publications and standards in the field. The number of motile events recorded per movie not readily predictable in advance of the experiment. |
| Data exclusions | For all single molecule measurements events longer than 4 consecutive pixels were included in the analysis. Microtubules not fully enclosed in the field of view or overlapping with another microtubule were excluded from the analysis. |
| Replication | Experiments were carried out as technical replicates, n=3. All attempts were successful. |
| Randomization | Datasets were not randomized. No group allocation was performed for any experiments. |
| Blinding | Blinding was not used. No group allocation was performed for any experiments. |

# Reporting for specific materials, systems and methods

We require information from authors about some types of materials, experimental systems and methods used in many studies. Here, indicate whether each material, system or method listed is relevant to your study. If you are not sure if a list item applies to your research, read the appropriate section before selecting a response.

### Materials & experimental systems

| n/a | Involved in the study |
|---|---|
| ☐ | ☒ Antibodies |
| ☐ | ☒ Eukaryotic cell lines |
| ☒ | ☐ Palaeontology and archaeology |
| ☒ | ☐ Animals and other organisms |
| ☒ | ☐ Clinical data |
| ☒ | ☐ Dual use research of concern |
| ☒ | ☐ Plants |

### Methods

| n/a | Involved in the study |
|---|---|
| ☒ | ☐ ChIP-seq |
| ☒ | ☐ Flow cytometry |
| ☒ | ☐ MRI-based neuroimaging |

# Antibodies

| | |
|---|---|
| Antibodies used | Primary antibodies: ANTI-FLAG® M2 mouse antibody (1:1000; Sigma- Aldrich F1804), anti-GAPDH (1:10,000; Cell Signaling Technology 2118), anti-acetylated tubulin (1:2000; Sigma-Aldrich T6793), anti-gamma-tubulin (1:5000; Sigma-Aldrich T6557). |

Secondary antibodies: Goat Anti-Mouse IgG StarBright Blue 700 (1:1000; BioRad 12004159), Goat Anti-Rabbit IgG (H + L) Alexa Fluor 647 (1:1000; Invitrogen A-21244).

Validation

Purity, specificity and sensitivity tests carried out by manufacturer. Antibodies widely used in the field.
ANTI-FLAG® M2 mouse antibody: two major bands with purity >90% when analyzed by microfluidic gel capillary electrophoresis; detects a single band of protein on a Western Blot from mammalian crude cell lysates; detects 2 ng of FLAG-BAP fusion protein by Dot Blot using Chemiluminescent Detection.
anti-GAPDH: GAPDH (14C10) Rabbit mAb detects endogenous levels of total GAPDH protein.

## Eukaryotic cell lines

Policy information about cell lines and Sex and Gender in Research

Cell line source(s)

Mouse IMCD-3-FlpIn cells (gift from Peter K. Jackson, Stanford; Mukhopadhyay et al (2010) Genes Dev. 24, 2180–2193). Sf9 cells (Gibco 11496015).

Authentication

Cell lines were not authenticated.

Mycoplasma contamination

Parental cell lines were verified to be free from mycoplasma using a PCR detection kit (Sigma-Aldrich, MP0035).

Commonly misidentified lines
(See ICLAC register)

No commonly misidentified lines used.

## Plants

Seed stocks

*Report on the source of all seed stocks or other plant material used. If applicable, state the seed stock centre and catalogue number. If plant specimens were collected from the field, describe the collection location, date and sampling procedures.*

Novel plant genotypes

*Describe the methods by which all novel plant genotypes were produced. This includes those generated by transgenic approaches, gene editing, chemical/radiation-based mutagenesis and hybridization. For transgenic lines, describe the transformation method, the number of independent lines analyzed and the generation upon which experiments were performed. For gene-edited lines, describe the editor used, the endogenous sequence targeted for editing, the targeting guide RNA sequence (if applicable) and how the editor was applied.*

Authentication

*Describe any authentication procedures for each seed stock used or novel genotype generated. Describe any experiments used to assess the effect of a mutation and, where applicable, how potential secondary effects (e.g. second site T-DNA insertions, mosiacism, off-target gene editing) were examined.*

