## [Peer Review File · Nature Structural & Molecular Biology]

Regulation of kinesin-2 motility by its beta-hairpin motif

Corresponding Author: Dr Anthony Roberts

Version 0:

Decision Letter:

31st Oct 2024

Dear Dr. Roberts,

Thank you for submitting your manuscript "Beta-hairpin Mechanism of Autoinhibition and Activation in the Kinesin-2 Family". I apologize for the delay in this decision

I am re-opening the manuscript submission link for you to resubmit your manuscript with all the associated files needed for the peer review process directly to our system, at your convenience. Please see below for details regarding the required materials. Please follow the link at the bottom of this email to upload the documents.

We want to ensure that the methods and statistics reporting in our papers are of the highest quality. To that end, we ask authors to fill out a Reporting Summary that collects information on experimental design and reagents, as well as an editorial Policy Checklist, which confirms compliance with our editorial policies, including the declaration of Competing Interests. If your paper includes ChIP-seq, flow cytometry or MRI data, we ask you take special care to complete those sections of the Reporting Summary as this data will aid greatly in the review of your manuscript.

These documents can be found by following the links below:

Reporting Summary:
<https://www.nature.com/documents/nr-reporting-summary.pdf>

Editorial Policy Checklist: <https://www.nature.com/documents/nr-editorial-policy-checklist.pdf>

Please be aware of our guidelines on digital image standards.

*****IMPORTANT*****

In order for us to proceed with the peer review process of your manuscript, we require you to provide accession numbers and reviewer tokens to access sequencing data sets. Please add this information to your manuscript file.

Please note we require official wwPDB validation reports for newly described atomic structures, as noted in the policy checklist.

We also request that authors provide cryo-EM maps, half-maps and models, to help the reviewers in assessing the work. We recommend the use of figshare integration into our systems, which allows for provision of anonymous access links for the referees (<https://www.springernature.com/gp/authors/research-data/figshare-integration>).

Alternatively, please upload .zip folders directly with the submission. To ensure the ease of reviewer access to the data, please specify in the Data Availability section, where the files can be found (provide a figshare link or direct the reader to the manuscript files).

Additionally, I would like to kindly request that you provide the code used to analyse the data to the reviewers, if used. In order for the reviewers to evaluate the work adequately they must be able to test the software/review the code themselves. If you have not yet provided the software, we therefore request that you provide a single compressed zip file containing the software with a readme.txt file or other user manual containing complete instructions for installing and running the software. If appropriate, please also provide example data and expected output. Sufficient material should be provided for referees to directly test the performance of the software/algorithm. If the software and materials are small enough to fit in a single compressed zip file less than 6MB in size, you may email this file directly to me. If the zip file is between 6 MB and 200 MB

you may upload it to our file transfer site. If necessary, a second zip file up to 200 MB in size can be used to supply the example data. Please let me know if you need to use this option and I'll send you further details. Alternatively, you can also upload the code to GitHub and provide us with the link.

Please also fill out and return to me the code and software submission checklist that will be made available to editors and reviewers during manuscript assessment. Please note that this form is a dynamic 'smart pdf' and must therefore be downloaded and completed in Adobe Reader, instead of opening it in a web browser.

<https://www.nature.com/documents/nr-software-policy.pdf>

Please use the link below to submit the files. **Please also remember to move forward all other files associated with this version of the paper.**

Link Redacted

Sincerely,
Kat

Katarzyna Ciazynska, PhD
(she/her)
Senior Editor
Nature Structural & Molecular Biology
<https://orcid.org/0000-0002-9899-2428>

Version 1:

Decision Letter:

9th Jan 2025

Dear Dr Roberts,

Thank you again for submitting your manuscript "Beta-hairpin Mechanism of Autoinhibition and Activation in the Kinesin-2 Family". I apologize for the delay in responding, which resulted from difficulty in obtaining the reviewer reports during the winter holidays. Nevertheless, we now have comments (below) from the 4 reviewers who evaluated your paper. In light of those reports, we remain interested in your study and would like to see your response to the comments of the referees, in the form of a revised manuscript.

You will see that while reviewers appreciate the results, they raise several concerns which will need to be addressed in a revision. Specifically, please ensure to validate the conclusions with further experiments where requested, particularly by referee #1 and #2. This includes direct evidence for the role of negatively charged residues in beta-hairpin motif, further mutagenesis and quantification to characterise the interfaces, characterisation of Kap3 (including controls requested by referee #1) and further support for the structural mimicry between tubulin and the beta-hairpin motif. We would like to stress that we find providing further experimental support essential for further consideration at NSMB.

Please be sure to address/respond to all concerns of the referees in full in a point-by-point response and highlight all changes in the revised manuscript text file.

We appreciate the requested revisions are extensive. We thus expect to see your revised manuscript within 6 months. If you cannot send it within this time, please let us know. We will be happy to consider your revision as long as nothing similar has been accepted for publication at NSMB or published elsewhere. Should your manuscript be substantially delayed without notifying us in advance and your article is eventually published, the received date would be that of the revised, not the original, version.

As you already know, we put great emphasis on ensuring that the methods and statistics reported in our papers are correct and accurate. As such, if there are any changes that should be reported, please submit an updated version of the Reporting

Summary along with your revision.

Reporting Summary:

EXTENDED DATA FIGURES

Please note that all key data shown in the main figures as cropped gels or blots should be presented in uncropped form, with molecular weight markers. These data can be aggregated into a single supplementary figure. While these data can be displayed in a relatively informal style, they must refer back to the relevant figures. These data should be submitted with the last revision, prior to acceptance, but you may want to start putting it together at this point.

We require deposition of coordinates (and, in the case of crystal structures, structure factors) into the Protein Data Bank with the designation of immediate release upon publication (HPUB). Electron microscopy-derived density maps and coordinate data must be deposited in EMDB and released upon publication. Deposition and immediate release of NMR chemical shift assignments are highly encouraged. Deposition of deep sequencing and microarray data is mandatory, and the datasets must be released prior to or upon publication. To avoid delays in publication, dataset accession numbers must be supplied with the final accepted manuscript and appropriate release dates must be indicated at the galley proof stage. Please find the complete NRG policies on data availability at <http://www.nature.com/authors/policies/availability.html>.

Link Redacted

Sincerely,
Kat

Katarzyna Ciazynska, PhD
(she/her)
Senior Editor
Nature Structural & Molecular Biology
<https://orcid.org/0000-0002-9899-2428>

Reviewers' Comments:

Reviewer #1 (Remarks to the Author):

Remarks to the Authors

This manuscript describes an interesting study of the molecular mechanisms of autoinhibition and activation of kinesin-2 motors, a heterotrimeric kinesin family whose regulation is poorly understood to date, as no autoinhibitory motif had been identified. Using structural predictions, cell biology and functional single-molecule assays, the authors describe a novel structural motif, the beta-hairpin, that autoinhibits kinesin-2 by preventing its binding to microtubules. They also provide a model for relief from this autoinhibition by the cargo adapter APC.

The regulation of motors, especially in bidirectional transport complexes, containing dynein and kinesin, is a major interest of the motor protein field. For this, a better understanding of kinesin autoinhibition is necessary, which is in part hampered by the paucity of structural information for full-length kinesin. This structural information is of major interest to the field, as exemplified by several recent studies about kinesin-1 regulation. Thus, this study is likely to be of interest to the motor protein and cytoskeleton field, as well as to the RNA transport field, given that APC is an RNA adaptor.

However, some of the conclusions of the paper are derived from structural predictions with very limited experimental validation, and some mechanistic details remain unexplored. Also, more information about the experimental details is necessary to judge the quality of the work. I would be supportive of a publication in NSMB if the authors could address these concerns and clarify open questions regarding their experimental work, as detailed below.

Major concerns:

1. In this study, the authors predict a beta-hairpin motif with negatively charged residues at its apex to be responsible for kinesin-2 autoinhibition. They provide evidence for the importance of the negatively charged residues in this motif. However, there is no direct evidence that these residues need to be in the context of the described beta-hairpin motif rather than functioning as a linear peptide motif, like the IAK motif in kinesin-1. It is also conceivable that the beta-hairpin structure is induced only upon interaction. Any experimental validation of the presence of this structure in the kinesin-2 tail would strongly support the model presented and potentially provide mechanistically insightful information, e.g. measurements of secondary structure content in purified tail domain by CD or NMR.

2. Results presented in Figure 2:

For assessment of the single molecule experiments presented here, a more detailed description of the experimental design is necessary. Were the experiments with Kif3A WT, Δ C38 and Δ C59 also done in triplicate? Does the graph mean to show datapoints that are zero; if so, why not three? Was there not a single binding event in these experiments or how was this data point obtained?

Regarding the experiments with truncated and activated mutants: From how many different motility chambers and fields of view were their >100 events collected? Were all microtubules in a field of view analyzed and if not, how were microtubules chosen for analysis? Does $n = 86$ MTs mean 86 MTs were analyzed per experiment or in total per sample or in total for all samples? This information is necessary to judge the relevance of the results.

3. Results in Figure 4:

Figure 4 C: Why did the authors use an unlabeled Kap3 for the MT binding assay with ATP? Can they exclude that Kap3 could bind to MTs itself instead of co-localizing with Kif3AB? Their signals in the AMPPNP experiment (lower panel) do not perfectly overlap and the MT looks very decorated.

Figure 4G: An experiment with a deletion of interface 2 only would be more convincing of its role in the interaction. The double-deletion is a rather big truncation which removes almost the entire tail and could have plenty of other effects. The same is true for the interface 1 experiment. Can the authors do point mutations in the interfaces instead of these drastic deletions as more direct proof for their interfaces? While the provided nsEM data indicate that the predicted overall fold of the complex is correct, they do not provide high resolution information which would be necessary to confirm interfaces precisely. In addition, a more quantitative analysis of these interactions would be informative and could disentangle the contributions of each surface and clarify whether they act synergistically, for example KD measurements (e.g. by SPR, MST or similar).

4. Results in Figure 5:

Figure 5 B,C: While this study provides evidence for an interaction between the beta-hairpin motif and the motor domain, there is no experimental validation of the second surface between each motor domain and its corresponding coiled-coil domain. By testing mutants in this second interface in their single molecule assay, the authors could provide much more convincing evidence for their hypothesis. Also, the individual contribution of each interaction (motor-tail vs. motor-stalk) could be tested.

5. Results in Figure 6:

Figure 6A: Can the authors please provide details about the APCARM mutant used in these experiments? How many conserved residues were mutated? Which were the aa identities in both WT and mutant? How did the authors ensure that this mutant protein retained correct folding? Without this information, the results presented here cannot meaningfully be interpreted.

Minor concerns:

1. Regarding results described in Figure 1:

The authors describe a structural motif derived from many AF2 and AF3 predictions. However, very little information about how these structures were predicted is provided. Were full-length sequences of all proteins used? Was the overall architecture of the complex reasonably predicted and if not, how did the authors interpret the results? Could the authors show (representative examples of) what an entire model looked like instead of features thereof only?

2. In the chapter "Impact of disrupting the β -hairpin on Kif3 cellular function", the authors call "autoinhibition via the β -hairpin motif crucial for Kif3 spatial regulation and coordination with dynein-2."

Could they please clarify what they mean by coordination with dynein-2? This is a bold statement given that there is no evidence provided for how dynein is affected (e.g. where is dynein localized? Does it accumulate at the ciliary tip?). In my opinion, the term coordination implies that the two motors are in the same complexes and dynein-2 probably gets stalled due to a tug-of-war. While this seems likely, an alternative explanation of for the observed Kif3B accumulation at the ciliary tip could be that it cannot interact with dynein complexes anymore.

3. Could the authors comment on the images presented in Figure S2C? Only some cells in the images shown for Kif3B WT and the KKK mutant show restored cilia, whereas others show Ac-tub signal accumulation similar to the KO. Is this a technical issue or potentially meaningful?

4. Figure 4E: What is the white part of the model? Is this the coiled-coil domain? Is this the same model as shown in Figure S5?

5. Figure S3D is referenced out of order. A description of the Kif3AC model and experiment should happen already in the results part of the manuscript and not only be mentioned at the end of the discussion.

Reviewer #2 (Remarks to the Author):

This important study by Webb and colleagues reveal the first molecular insights into the transport kinesin-2 motor family the molecular mechanism of autoinhibition and propose a model for activation of kinesin-2 motors. Kinesin-2 are processive motors essential for cellular transport. They associate as trimers with Kif3a, Kif3b and KAP3, but the molecular basis is unknown. Here the authors identify a structural feature in the C terminus of Kinesin-2 that stabilize the autoinhibited state using an integrated structural approach. The KAP3 protein associates with Kif3a and Kif3b, but the complex remains non motile.

The beta-hairpin structure is highly conserved through species. They show that removing the beta-hairpin of Kif3a activated the motor, and when both the beta-hairpins of Kif3a and b were removed, the landing rate increased. They then use mouse cells that grow cilia. There, Kif3 complexes transport IFT trains to the tip. Using CRISPR-cas9, the cilia fail to grow in the absence of Kif3b, but cilia growth is rescued by WT Kif3b expression. A beta-hairpin Kif3b mutant allows growth of the cilia, however it accumulates at its end rather than being transported back by Dynein-2 to the base, suggesting auto-inhibition is key for its return to the cell body and dynein-mediated transport.

The authors then reconstitute a Kif3a/Kif3b/KAP complex, present class averages and an AlphaFold model of Kif3a/Kif3b/KAP, showing KAP3 binds the unstructured tails of Kif3a and b and part of the hairpin. It seems the class averages stabilize an autoinhibition of the complex. In their last set of biochemical reconstitution, the authors generate a complex of Kif3a/Kif3b/KAP bound to Armadillo domain of APC, known to bind this complex. They show that in the presence of APC, Kif3a/Kif3b/KAP is a motile complex, but an APC mutant (mutations guided by AlphaFold prediction of interaction) does not.

When structures generated by AlphaFold are presented, the authors should make it clear in the figure panel. Because there are also some figures with EM data, it can lead the reader to believe the structural models are derived from EM data. I am not convinced the data support a molecular mimicry model, where the beta-hairpin and tubulin compete for binding as it stands. If the authors want to propose this, further experiments are necessary. Alternatively, editing the text to remove the molecular mimicry message would be fine. Overall, the paper is very well-written, the data are compelling and the figures are well-designed. The data largely support the conclusions. I am enthusiastic about its publication after revision.

Major points

1. More cellular characterisation of KAP3 in the mammalian cells would strengthen the manuscript and conclusions. What happens in the mouse cells, when KAP is absent? Is Kif3A/b at the base and there is no cilia growth?

Where is KAP3 in the presence of the various Kif3b mutants in ciliated cells? I would have liked to see more characterisation of KAP3 to understand where it is in cells and what it does to understand its effect on kif3a/b.

2. If APC activates Kif3a/b/KAP3 (Fig 4C), does the complex have an extended conformation (Fig 6D)? Can the authors tell from the SEC or negative stain images?

2. What happens when the beta-hairpin of Kif3b is removed in the context of full-length kif3a?

3. There is not enough data to support molecular mimicry of the beta-hairpin, because the hairpin is structured, but the C termini of tubulin much less. If the authors want to propose this, crosslinking the motor to the beta-hairpin or titration experiments varying the amount of hairpin added to motors in the presence of microtubules would support this (as for Kinesin-1 and Kinesin-13). They could also do experiments in the presence of microtubules or mutate the motor at the sites identified in figure 5. Alternatively, editing the text to remove the molecular mimicry message would work.

Minor points

1. Introduction. Could give more background on kinesin 2 function at the start. Currently this information is in the first paragraph of the discussion.

2. Can you comment on the affinity of KAP for Kif3a/b apart from the nanomolar affinity? Are they obligate interactors in the cell?

3. What is the behavior of kif3c? Do we expect the same properties? Does it have different properties or expression in cells?

4. Page 5. First sentence, it would be good to say the mutations are in the context of the full-length motor.

5. Figure 2. The diagrams are slightly confusing because the experiments are on the construct containing the motor domain. In figure 2c, they could show a close up on the full-length motor with the hairpin, with exactly the same schematics.

6. When structures generated by AlphaFold are presented, the authors should make it clear in the figure panel. Because there are also some figures with EM data, it can lead the reader to believe the structural models are derived from EM data.

Reviewer #3 (Remarks to the Author):

Webb and coauthors employ a host of approaches (biochemistry, in vitro motility assays, immunofluorescence of fixed cells, and negative-stain electron microscopy) to characterize the molecular mechanism of kinesin-2 autoinhibition. They show the beta-hairpin motif in the tail of kinesin-2s mediates autoinhibition. Further, autoinhibition can occur when the kinesin-2 adaptor KAP3 is bound, where KAP3 instead forms an interface for interactions with other cargo adaptors.

Kinesin-2s drive intraflagellar transport and axonal transport of vesicular cargoes and mRNA. While the autoinhibition mechanism of kinesin-1 has been elucidated in ever greater detail, relatively little was known about how kinesin-2's activity is regulated. This manuscript makes an important contribution by filling this gap. Experiments were performed carefully, analyzed with rigor, and interpreted appropriately.

Despite my enthusiasm for the manuscript, there are several points that could be strengthened:

- The electron microscopy and alpha fold predictions indicate an autoinhibited KIF3AB-Kap3 structure, with the motor domains in close proximity to the beta-hairpins. Is there also evidence of an active, unfolded conformation in any of the class averages? What fraction of particles are represented in the class averages presented?

- While the data make it clear that the beta-hairpin is sufficient to autoinhibit kinesin-2, do the authors expect other interactions between the coiled-coil regions of kinesin-2, similar to recent results for kinesin-1 (Cianfracco and McKenney labs), contribute to stabilizing the autoinhibited conformation?

- Are there specific interactions between the beta-hairpins and APC that would exclude autoinhibition when bound to an adaptor? Can these data provide insight into cases where kinesin-2 is expected to be bound to a cargo adaptor but in an autoinhibited state, for example in IFT and melanosome transport?

- The descriptions of methods are overly reliant on references to previous work. Please include adequate details to assess the results (passivation, kymograph analysis).

- The run lengths for the deltaC156 construct are ~50% lower than reported in Guzik-Lendrum et al., 2015, Biophys. J.), although the results are consistent with Andreasson et al., 2015, Current Biology. There is a large variance in the run lengths for the deltaC156 construct between the three experiments. Can your experiments provide any insight into the cause of these discrepant results between labs and the variance observed in these results?

- Run lengths are similar to the resolution of diffraction-limited images. What is the expected resolution of the analysis used here?

- Analysis of the distributions of run lengths and velocities might provide insight into the co-existence of activated/inhibited states at different conditions / truncations.

- In Fig. 3, the effects of the different KIF3B constructs may result from variable expression levels of the constructs. These results would be strengthened by quantification of the protein levels of the KIF3B constructs in the cells compared. Do the KIF3B mutant constructs localize differently as would be expected?

Minor comments:

- What is the rationale for using different buffers for the APC experiments than the other assays?
- Lack of statistical comparisons (Fig. 2, 4G)
- Include a map of mutations in the APC-mut construct (Fig. 6).
- List the number of kymographs used for analysis (Fig. 2C)
- It would be instructive to include quantification of the mutant constructs in Fig. 2D

Reviewer #4 (Remarks to the Author):

In their manuscript, Webb and coauthors use a combination of approaches (EM, AlphaFold, single and multimolecule motility assays, biochemical analysis, cell biology) to identify the structural basis of the auto inhibition mechanisms of kinesin-2s. The results are novel and, in my mind very important. Kinesin-2's are key motor proteins the on and off switching of which is of utmost relevance to many cellular processes. Data and interpretation are extremely well presented and very convincing. The manuscript is very well written, also highly accessible to a broader audience. The manuscript fits well in the context of research performed on this topic before and the previous literature is well referenced.

I only have one key question to the authors. If that one is answered satisfactorily I think this manuscript should be published in NSMB, I think it is one of the best and most exciting manuscripts I have read this year!

My question is regarding the acidic residues in the KIF3A beta-hairpin. In figure 1A they are E615 and D616, while in the context of the autoinhibition (figure 5B, p9 I8) they are indicated as E612 and D613. Why are these numbers different? I hope this a simple miss labeling!

In their revised version, the authors might consider the following textual suggestions (small points):

- p6 I11/12 I think it should be line (without 'S') scans.
- p6 I15 "The Kif3B beta-hairpin mutant" it would have helped me if you would have indicated also as "KKK" mutant (to connect with before and figure)
- in the discussion I was hoping for a short comparison of the kinesin-1 and kinesin-2 mechanisms and the reason / consequence why they are different.

Erwin J.G. Peterman

Version 2:

Decision Letter:

Our ref: NSMB-A49925B

17th Apr 2025

Dear Dr. Roberts,

Thank you for submitting your revised manuscript "Beta-hairpin Mechanism of Autoinhibition and Activation in the Kinesin-2 Family" (NSMB-A49925B). It has now been seen by the original referees and their comments are below.

You will see that reviewer #1 commented on your responses to reviewer #2 comments. We asked them for this additional input, since reviewer #2 was not available to assess this revision.

The reviewers find that the paper has improved in revision, and therefore we'll be happy in principle to publish it in Nature Structural & Molecular Biology, pending minor revisions to satisfy the referees' final requests and to comply with our editorial and formatting guidelines.

We are now performing detailed checks on your paper and will send you a checklist detailing our editorial and formatting requirements in about 2-3 weeks. Please do not upload the final materials and make any revisions until you receive this additional information from us.

To facilitate our work at this stage, it is important that we have a copy of the main text as a word file. If you could please send along a word version of this file as soon as possible, we would greatly appreciate it; please make sure to copy the NSMB account (cc'ed above).

Sincerely,
Kat

Katarzyna Ciazynska, PhD
(she/her)
Senior Editor

Reviewer #1 (Remarks to the Author):

This manuscript describes an interesting study of the molecular mechanisms of autoinhibition and activation of kinesin-2 motors, a heterotrimeric kinesin family whose regulation is poorly understood to date, as no autoinhibitory motif had been identified. Using structural predictions, cell biology and functional single-molecule assays, the authors describe a novel structural motif, the beta-hairpin, that autoinhibits kinesin-2 by preventing its binding to microtubules. They also provide a model for relief from this autoinhibition by the cargo adapter APC.

Whereas in the initial manuscript, several conclusions were based heavily on predictions, the authors now provide additional experimental validation of their model, strengthening the claims of the manuscript. Also, experimental details necessary for understanding and judging the work have been clarified. While I still think that the dissection of the different interfaces of the Kif3AB-Kap3 interaction would be valuable, I understand the authors' reasoning against showing these experiments.

I am now supportive of a publication in NSMB.

****additional comments****

I have worked through Reviewer 2's comments point by point and checked how the authors responded to them. In summary, I think all points have been addressed sufficiently, either experimentally or with reasonable explanations from literature or of the technical limitations preventing experiments (point 2). These might be worth mentioning in the manuscript, too, as other readers might wonder as well.

Here are my comments point by point:

This important study by Webb and colleagues reveal the first molecular insights into the transport kinesin-2 motor family the molecular mechanism of autoinhibition and propose a model for activation of kinesin-2 motors. Kinesin-2 are processive motors essential for cellular transport. They associate as trimers with Kif3a, Kif3b and KAP3, but the molecular basis is unknown. Here the authors identify a structural feature in the C terminus of Kinesin-2 that stabilize the autoinhibited state using an integrated structural approach. The KAP3 protein associates with Kif3a and Kif3b, but the complex remains non motile.

The beta-hairpin structure is highly conserved through species. They show that removing the beta-hairpin of Kif3a activated the motor, and when both the beta-hairpins of Kif3a and b were removed, the landing rate increased. They then use mouse cells that grow cilia. There, Kif3 complexes transport IFT trains to the tip. Using CRISPR-cas9, the cilia fail to grow in the absence of Kif3b, but cilia growth is rescued by WT Kif3b expression. A beta-hairpin Kif3b mutant allows growth of the cilia, however it accumulates at its end rather than being transported back by Dynein-2 to the base, suggesting auto-inhibition is key for its return to the cell body and dynein-mediated transport. The authors then reconstitute a Kif3a/Kif3b/KAP complex, present class averages and an AlphaFold model of Kif3a/Kif3b/KAP, showing KAP3 binds the unstructured tails of Kif3a and b and part of the hairpin. It seems the class averages stabilize an autoinhibition of the complex. In their last set of biochemical reconstitution, the authors generate a complex of Kif3a/Kif3b/KAP bound to Armadillo domain of APC, known to bind this complex. They show that in the presence of APC, Kif3a/Kif3b/KAP is a motile complex, but an APC mutant (mutations guided by AlphaFold prediction of interaction) does not.

When structures generated by AlphaFold are presented, the authors should make it clear in the figure panel. Because there are also some figures with EM data, it can lead the reader to believe the structural models are derived from EM data. I am not convinced the data support a molecular mimicry model, where the beta-hairpin and tubulin compete for binding as it stands. If the authors want to propose this, further experiments are necessary. Alternatively, editing the text to remove the molecular mimicry message would be fine. Overall, the paper is very well-written, the data are compelling and the figures are well-designed. The data largely support the conclusions. I am enthusiastic about its publication after revision.

Major points

1. More cellular characterisation of KAP3 in the mammalian cells would strengthen the manuscript and conclusions. What happens in the mouse cells, when KAP is absent? Is Kif3A/b at the base and there is no cilia growth

Where is KAP3 in the presence of the various Kif3b mutants in ciliated cells? I would have liked to see more characterisation of KAP3 to understand where it is in cells and what it does to understand its effect on kif3a/b.

Thank you for this suggestion. We generated new stable cell lines to assess the localization of Kap3 in the Kif3B WT and KKK mutant backgrounds (Extended Data Fig. 2e). We found that whereas Kap3 is distributed along the length of the cilium in the background of Kif3B

WT, it is strongly accumulated at the ciliary tip in the Kif3B KKK mutant, mirroring the localization of the Kif3B subunit. A previous study investigated the impact of Kap3 deletion in mouse. Similar to Kif3A or Kif3B KO, Kap3 KO results in embryonic lethality, presumably due to the loss of cilia (PMID: 15834408). Loss of cilia is observed in a Chlamydomonas KAP temperature sensitive mutant at the restrictive temperature ((PMID: 15616187). This study also shows that the kinesin-2 complex is not efficiently targeted to the basal body region at the KAP restrictive temperature.

>The authors address the question of Kap3 localization experimentally by assessing it in the Kif3B KKK mutant compared to Kif3B WT, answering the reviewer's question.

The question about the consequences of lack of Kap3 was previously answered, as the authors point out by referencing the relevant literature.

52. If APC activates Kif3a/b/KAP3 (Fig 4C), does the complex have an extended conformation (Fig 6D)? Can the authors tell from the SEC or negative stain images?

To address this interesting point, we used negative stain EM to investigate the architecture of Kif3AB-Kap3 bound to APCARM

. As shown in Extended Data Fig. 4d, Kif3AB-Kap3-

APCARM often has an extended appearance, in contrast to Kif3AB-Kap3 without APCARM

, which is compact (Extended Data Fig. 3b).

>The data requested by the reviewer is shown in Extended Data Fig. 4.

2. What happens when the beta-hairpin of Kif3b is removed in the context of full-length kif3a?

We have also been curious about this question. We found that (unlike other constructs) Kif3AB aggregates when the Kif3B tail is removed and the Kif3A tail is left intact, meaning that an analysis of motility is not possible for this construct.

>If the requested mutant is indeed insoluble, this is a technical limitation that cannot easily be overcome and it is acceptable that the authors state this. It could be worth mentioning in the manuscript that this was tested.

3. There is not enough data to support molecular mimicry of the beta-hairpin, because the hairpin is structured, but the C termini of tubulin much less. If the authors want to propose this, crosslinking the motor to the beta-hairpin or titration experiments varying the amount of hairpin added to motors in the presence of microtubules would support this (as for Kinesin-1 and Kinesin-13). They could also do experiments in the presence of microtubules or mutate the motor at the sites identified in figure 5. Alternatively, editing the text to remove the molecular mimicry message would work.

We think there has been a misunderstanding here, which has prompted us to clarify our language. We did not mean that the β -hairpin motif mimics the disordered C-terminal tails of tubulin. Rather, we meant that the residues in the Kif3AB motor domains involved in the β -hairpin motif and C-terminal coiled coil interaction are the same as those used to bind the globular core of tubulin. In this sense, the negative charged surface of the β -hairpin motif and C-terminal coiled coil mimic the negatively charged surface of the tubulin globular core. We have edited the text, which we hope now avoids this ambiguity.

>I agree with the authors that the text has been edited sufficiently to clarify this point. The new wording is more cautious and avoids overinterpretation.

Minor points

1. Introduction. Could give more background on kinesin 2 function at the start. Currently this information is in the first paragraph of the discussion.

We have elaborated on kinesin-2 function in the Introduction.

2. Can you comment on the affinity of KAP for Kif3a/b apart from the nanomolar affinity? Are they obligate interactors in the cell?

Our new cellular localization experiments show that Kap3 localization closely matches that of Kif3B, in agreement with a tight interaction between them in the cell. However, it is not possible to say that Kif3AB never exists without Kap3 bound i.e. they are true obligate interactors. For example, as highlighted by Scholey (2013), when Kif3 is purified from cells, the molar ratio of the Kap3 subunit is usually estimated to be slightly less than one (e.g. 1:1:0.8 in the case of the KifA:Kif3B:Kap3 orthologs in *C. reinhardtii*) (PMID: 23750925).

3. What is the behavior of kif3c? Do we expect the same properties? Does it have different properties or expression in cells?

As shown in Extended Data Fig. 3, Kif3AC-Kap3 is predicted to adopt a similar autoinhibited architecture to Kif3AB-Kap3. Our single-molecule binding experiment (Extended Data Fig. 3d) shows that Kif3AC binds to Kap3 in a tail-dependent manner, similar to Kif3AB. Interestingly, however, while Kif3AC is predicted to use Interface 1 and 2 to bind Kap3, in a similar fashion to Kif3AB, Kif3AC is predicted to lack Interface 3 with Kap3 (Extended Data Fig. 3c). In vivo, Kif3AC's role is different to Kif3AB, as it does not function in intraflagellar transport and is restricted to cytoplasmic transport. Thus, Interface 3, which involves the C-terminal region of Kif3B contributing a β -strand and α -helix to a β -sheet at the Kap3 N-terminus, may dictate Kif3AB specificity for intraflagellar transport.

4. Page 5. First sentence, it would be good to say the mutations are in the context of the full-length motor.

Amended.

5. Figure 2. The diagrams are slightly confusing because the experiments are on the construct containing the motor domain. In figure 2c, they could show a close up on the full-length motor with the hairpin, with exactly the same schematics.

Thank you for this suggestion. We have modified the schematic to make it clear that the tail diagrams represent a closeup of a larger construct.

6. When structures generated by AlphaFold are presented, the authors should make it clear in the figure panel. Because there are also some figures with EM data, it can lead the reader to believe the structural models are derived from EM data.

We have emphasized in the figure legends which panels show AlphaFold3 models and which show EM data. We also now write 'AlphaFold3' in full, rather than using the 'AF3' abbreviation, to further clarify this point.

7

>Reviewer 2's minor concerns have all been addressed as requested.

Reviewer #3 (Remarks to the Author):

The authors addressed all of my concerns through new experiments, analysis, and clarifications to the text. In brief, they:

- Performed negative-stain EM on the Kif3AB-KAP3-APC(ARM) complex (Fig. S3)
- Generated stable cell lines to examine the effects of Kif3B mutations on the localization of Kap3.
- Compared the autoinhibition mechanisms of Kif5 and Kif3 in the discussion
- Added details to the methods section as requested
- Provided statistical analysis to results in Figs. 2 and 4.

The manuscript provides important new insights into the autoinhibitory mechanisms for kinesin-2 and its regulation by adaptor proteins.

We are grateful to the Reviewers for their positive comments and suggestions to improve the manuscript. To address their comments, we have generated new constructs and cell lines, performed new experiments and analysis, and modified the figures and text, as detailed in the point-by-point response below.

Reviewer #1 (Remarks to the Author):

This manuscript describes an interesting study of the molecular mechanisms of autoinhibition and activation of kinesin-2 motors, a heterotrimeric kinesin family whose regulation is poorly understood to date, as no autoinhibitory motif had been identified. Using structural predictions, cell biology and functional single-molecule assays, the authors describe a novel structural motif, the beta-hairpin, that autoinhibits kinesin-2 by preventing its binding to microtubules. They also provide a model for relief from this autoinhibition by the cargo adapter APC.

The regulation of motors, especially in bidirectional transport complexes, containing dynein and kinesin, is a major interest of the motor protein field. For this, a better understanding of kinesin autoinhibition is necessary, which is in part hampered by the paucity of structural information for full-length kinesin. This structural information is of major interest to the field, as exemplified by several recent studies about kinesin-1 regulation. Thus, this study is likely to be of interest to the motor protein and cytoskeleton field, as well as to the RNA transport field, given that APC is an RNA adaptor.

However, some of the conclusions of the paper are derived from structural predictions with very limited experimental validation, and some mechanistic details remain unexplored. Also, more information about the experimental details is necessary to judge the quality of the work. I would be supportive of a publication in NSMB if the authors could address these concerns and clarify open questions regarding their experimental work, as detailed below.

Major concerns:

1. In this study, the authors predict a beta-hairpin motif with negatively charged residues at its apex to be responsible for kinesin-2 autoinhibition. They provide evidence for the importance of the negatively charged residues in this motif. However, there is no direct evidence that these residues need to be in the context of the described beta-hairpin motif rather than functioning as a linear peptide motif, like the IAK motif in kinesin-1. It is also conceivable that the beta-hairpin structure is induced only upon interaction. Any experimental validation of the presence of this structure in the kinesin-2 tail would strongly support the model presented and potentially provide mechanistically insightful information, e.g. measurements of secondary structure content in purified tail domain by CD or NMR.

Thank you for this suggestion to strengthen the manuscript. We generated a new mutant to test if the negatively charged residues need to be in the context of the described β -hairpin motif or can function as a linear peptide motif. We introduced glycine, the residue with lowest β -strand propensity, into the strands flanking the negatively charged residues in Kif3A, while leaving the negatively charged residues themselves intact. An AlphaFold3 model predicts that these mutations disrupt β -hairpin formation. We performed motility assays with the glycine mutant and found that it is activated (new **Fig. 2d** and **Extended Data Table 1**), in contrast to the wild-type construct, and despite still containing the negatively charged residues. These experiments provide strong evidence that the negatively charged residues need to be in the context of an intact β -hairpin motif rather than functioning as a linear peptide motif. It is an interesting question whether the β -hairpin is pre-formed or induced upon interaction with the motor domains. A recent pre-print, submitted to bioRxiv following our study, investigates the structure of the kinesin-2 tail in a construct lacking the motor domains and majority of coiled

coil (<https://doi.org/10.1101/2025.03.21.644525>). The β -hairpin motif is found to be intact when the motor domains are artificially removed. This result suggests that the β -hairpin is pre-formed rather than induced upon interaction with the motor domains, providing further support for our model in **Fig. 5d**.

2. Results presented in Figure 2:

For assessment of the single molecule experiments presented here, a more detailed description of the experimental design is necessary. Were the experiments with Kif3A WT, Δ C38 and Δ C59 also done in triplicate? Does the graph mean to show datapoints that are zero; if so, why not three? Was there not a single binding event in these experiments or how was this data point obtained?

Regarding the experiments with truncated and activated mutants: From how many different motility chambers and fields of view were their >100 events collected? Were all microtubules in a field of view analyzed and if not, how were microtubules chosen for analysis? Does $n = 86$ MTs mean 86 MTs were analyzed per experiment or in total per sample or in total for all samples? This information is necessary to judge the relevance of the results.

Thank you for highlighting the need to clarify these points, which we have done in the figure legend and Methods and summarize here. Kif3A WT, Δ C38 and Δ C59 motility assay were performed in triplicate. No motile events were observed across the three experimental repeats for these constructs, suggesting they are strongly autoinhibited at the single-molecule level. We used a cross to denote this lack of motility rather than three conventional data points, in an effort to highlight that it is not that a velocity of 0 nm s^{-1} was measured three times but rather that there were no motile events; a subtle but important distinction that we have now explained in the figure legend. For experiments with truncated and activated mutants, three different motility chambers were analyzed for each construct, with three movies collected per motility chamber, giving a total of nine separate movies for each of the 15 constructs analyzed. Microtubules not fully enclosed in the field of view or overlapping with another microtubule were excluded from the analysis. For the microtubule gliding assay, $n = 86$ microtubules referred to 86 microtubules analyzed in total across the three separate experiments. We now give the number of microtubules per replicate in the figure legend to avoid this ambiguity.

3. Results in Figure 4:

Figure 4 C: Why did the authors use an unlabeled Kap3 for the MT binding assay with ATP? Can they exclude that Kap3 could bind to MTs itself instead of co-localizing with Kif3AB? Their signals in the AMPPNP experiment (lower panel) do not perfectly overlap and the MT looks very decorated.

Figure 4G: An experiment with a deletion of interface 2 only would be more convincing of its role in the interaction. The double-deletion is a rather big truncation which removes almost the entire tail and could have plenty of other effects. The same is true for the interface 1 experiment. Can the authors do point mutations in the interfaces instead of these drastic deletions as more direct proof for their interfaces? While the provided nsEM data indicate that the predicted overall fold of the complex is correct, they do not provide high resolution information which would be necessary to confirm interfaces precisely.

In addition, a more quantitative analysis of these interactions would be informative and could disentangle the contributions of each surface and clarify whether they act synergistically, for example KD measurements (e.g. by SPR, MST or similar).

We performed the microtubule-binding assay in ATP conditions using labelled or unlabelled Kap3 and observed the same result. We have updated the upper panel of **Fig. 3c** to show the version of the experiment with labelled Kap3 for ease of comparison with the lower panel. This experiment shows that Kap3 does not bind to the microtubule independently of Kif3AB, in agreement with **Extended Data Fig. 3d**, which also used labelled Kap3. We also repeated

the AMPNP experiment with a sparser density of Kif3AB which shows the co-localization with Kap3 more clearly (new **Fig. 3c**, lower panel). We agree that further dissection of the Kif3AB-Kap3 interface will be interesting, especially as it is likely that the strength of Interface 2 is regulated by phosphorylation. We hope the Reviewer agrees that these affinity measurements are the basis of a separate manuscript on Kif3 phospho-regulation and lie beyond the scope of the current study.

4. Results in Figure 5:

Figure 5 B,C: While this study provides evidence for an interaction between the beta-hairpin motif and the motor domain, there is no experimental validation of the second surface between each motor domain and its corresponding coiled-coil domain. By testing mutants in this second interface in their single molecule assay, the authors could provide much more convincing evidence for their hypothesis. Also, the individual contribution of each interaction (motor-tail vs. motor-stalk) could be tested.

To probe the importance of the interaction between the Kif3AB motor domains and C-terminal coiled coil segment for autoinhibition, we mutated three negatively charged residues in the Kif3A coiled coil (E559K, D562K, E566K) and Kif3B coiled coil (D554K, D557K, E561K) at the interface with the autoinhibited motor domains. While the mutations had no effect on Kif3AB heterodimerization, they activated the motility of Kif3AB akin to the removal of the β -hairpin motif (**Fig. 5e**, **Extended Data Table 1**). We can now conclude that both the β -hairpin motif and C-terminal coiled coil segment are critical for Kif3AB autoinhibition, in full agreement with the proposed mechanism. Thank you for this suggestion.

5. Results in Figure 6:

Figure 6A: Can the authors please provide details about the APC^{ARM} mutant used in these experiments? How many conserved residues were mutated? Which were the aa identities in both WT and mutant? How did the authors ensure that this mutant protein retained correct folding? Without this information, the results presented here cannot meaningfully be interpreted.

We apologize this information was not sufficiently clear in the original submission. The mutated residues in APC^{ARM} are those shown as black spheres in **Fig. 6d**, and we now give the mutations explicitly in the figure legend. We verified that the APC^{ARM} mutant has the same elution volume (hydrodynamic radius) as the wild-type construct using size-exclusion chromatography (now shown in **Extended Data Fig. 4a**).

Minor concerns:

1. Regarding results described in Figure 1:

The authors describe a structural motif derived from many AF2 and AF3 predictions. However, very little information about how these structures were predicted is provided. Were full-length sequences of all proteins used? Was the overall architecture of the complex reasonably predicted and if not, how did the authors interpret the results? Could the authors show (representative examples of) what an entire model looked like instead of features thereof only?

Full-length sequences were used for AlphaFold predictions (now clarified in Methods). We have added the entire model of human Kif3AB to **Extended Data Fig. 1** colored by pLDDT score and with a PAE plot. We show the entire model of Kif3AB-Kap3 and Kif3AC-Kap3 in **Extended Data Fig. 3c,d**. Each complex is predicted with high (90 > pLDDT > 70) to very high (pLDDT > 90) confidence, with the expected exception of disordered segments.

2. In the chapter “Impact of disrupting the β -hairpin on Kif3 cellular function”, the authors call “autoinhibition via the β -hairpin motif crucial for Kif3 spatial regulation and coordination with dynein-2.”

Could they please clarify what they mean by coordination with dynein-2? This is a bold statement given that there is no evidence provided for how dynein is affected (e.g. where is dynein localized? Does it accumulate at the ciliary tip?). In my opinion, the term coordination implies that the two motors are in the same complexes and dynein-2 probably gets stalled due to a tug-of-war. While this seems likely, an alternative explanation of for the observed Kif3B accumulation at the ciliary tip could be that it cannot interact with dynein complexes anymore.

Thank you for highlighting our need to clarify our language here. We meant “coordination” in the general sense: to “bring the different elements of (a complex activity or organization) into a harmonious or efficient relationship”. This definition encompasses both interesting possibilities mentioned: an intact β -hairpin motif is required to prevent a tug-of-war with dynein-2, or an intact β -hairpin is required for Kif3 to associate with retrograde intraflagellar transport trains. To avoid ambiguity with other definitions of coordination, we have rephrased the highlighted passage. Related to this point, as well as showing that mutation of the β -hairpin motif causes Kif3B to accumulate at the ciliary tip and prevents its efficient recycling to the cell body, we have added a new experiment to show that Kap3 also accumulates at the ciliary tip in the background of this Kif3B β -hairpin motif mutant (see also response to Reviewer #2, point 1).

3. Could the authors comment on the images presented in Figure S2C? Only some cells in the images shown for Kif3B WT and the KKK mutant show restored cilia, whereas others show Ac-tub signal accumulation similar to the KO. Is this a technical issue or potentially meaningful?

As quantified in **Fig. 3a**, while there is intrinsic variability in cilia length (reflected in the images presented in **Extended Data Fig. 2c**, which represent a random sampling of the much larger number of cilia quantified), the mean cilia length is not statistically different for the control, Kif3B WT and KKK mutant backgrounds.

4. Figure 4E: What is the white part of the model? Is this the coiled-coil domain? Is this the same model as shown in Figure S5?

Thank you for flagging these unclear aspects of **Fig. 4E**. The white part of the model is the coiled coil domain and the model is the same as that shown in **Fig. S5**. We now clarify these points the figure legend.

5. Figure S3D is referenced out of order. A description of the Kif3AC model and experiment should happen already in the results part of the manuscript and not only be mentioned at the end of the discussion.

Thank you for this suggestion. We now refer to the Kif3AC experiment (**Extended Data Fig. 3d**) in the Results section.

Reviewer #2 (Remarks to the Author):

This important study by Webb and colleagues reveal the first molecular insights into the transport kinesin-2 motor family the molecular mechanism of autoinhibition and propose a model for activation of kinesin-2 motors. Kinesin-2 are processive motors essential for cellular transport. They associate as trimers with Kif3a, Kif3b and KAP3, but the molecular basis is unknown. Here the authors identify a structural feature in the C terminus of Kinesin-2 that stabilize the autoinhibited state using an integrated structural approach. The KAP3 protein associates with Kif3a and Kif3b, but the complex remains non motile.

The beta-hairpin structure is highly conserved through species. They show that removing the beta-hairpin of Kif3a activated the motor, and when both the beta-hairpins of Kif3a and b were removed, the landing rate increased. They then use mouse cells that grow cilia. There, Kif3 complexes transport IFT trains to the tip. Using CRISPR-cas9, the cilia fail to grow in the absence of Kif3b, but cilia growth is rescued by WT Kif3b expression. A beta-hairpin Kif3b mutant allows growth of the cilia, however it accumulates at its end rather than being transported back by Dynein-2 to the base, suggesting auto-inhibition is key for its return to the cell body and dynein-mediated transport. The authors then reconstitute a Kif3a/Kif3b/KAP complex, present class averages and an Alphafold model of Kif3a/Kif3b/KAP, showing KAP3 binds the unstructured tails of Kif3a and b and part of the hairpin. It seems the class averages stabilize an autoinhibition of the complex. In their last set of biochemical reconstitution, the authors generate a complex of Kif3a/Kif3b/KAP bound to Armadillo domain of APC, known to bind this complex. They show that in the presence of APC, Kif3a/Kif3b/KAP is a motile complex, but an APC mutant (mutations guided by Alphafold prediction of interaction) does not.

When structures generated by Alphafold are presented, the authors should make it clear in the figure panel. Because there are also some figures with EM data, it can lead the reader to believe the structural models are derived from EM data. I am not convinced the data support a molecular mimicry model, where the beta-hairpin and tubulin compete for binding as it stands. If the authors want to propose this, further experiments are necessary. Alternatively, editing the text to remove the molecular mimicry message would be fine. Overall, the paper is very well-written, the data are compelling and the figures are well-designed. The data largely support the conclusions. I am enthusiastic about its publication after revision.

Major points

1. More cellular characterisation of KAP3 in the mammalian cells would strengthen the manuscript and conclusions. What happens in the mouse cells, when KAP is absent? Is Kif3A/b at the base and there is no cilia growth

Where is KAP3 in the presence of the various Kif3b mutants in ciliated cells? I would have liked to see more characterisation of KAP3 to understand where it is in cells and what it does to understand its effect on kif3a/b.

Thank you for this suggestion. We generated new stable cell lines to assess the localization of Kap3 in the Kif3B WT and KKK mutant backgrounds (**Extended Data Fig. 2e**). We found that whereas Kap3 is distributed along the length of the cilium in the background of Kif3B WT, it is strongly accumulated at the ciliary tip in the Kif3B KKK mutant, mirroring the localization of the Kif3B subunit. A previous study investigated the impact of Kap3 deletion in mouse. Similar to Kif3A or Kif3B KO, Kap3 KO results in embryonic lethality, presumably due to the loss of cilia (PMID: 15834408). Loss of cilia is observed in a *Chlamydomonas* KAP temperature sensitive mutant at the restrictive temperature ((PMID: 15616187). This study also shows that the kinesin-2 complex is not efficiently targeted to the basal body region at the KAP restrictive temperature.

2. If APC activates Kif3a/b/KAP3 (Fig 4C), does the complex have an extended conformation (Fig 6D)? Can the authors tell from the SEC or negative stain images?

To address this interesting point, we used negative stain EM to investigate the architecture of Kif3AB-Kap3 bound to APC^{ARM}. As shown in **Extended Data Fig. 4d**, Kif3AB-Kap3-APC^{ARM} often has an extended appearance, in contrast to Kif3AB-Kap3 without APC^{ARM}, which is compact (**Extended Data Fig. 3b**).

2. What happens when the beta-hairpin of Kif3b is removed in the context of full-length kif3a?

We have also been curious about this question. We found that (unlike other constructs) Kif3AB aggregates when the Kif3B tail is removed and the Kif3A tail is left intact, meaning that an analysis of motility is not possible for this construct.

3. There is not enough data to support molecular mimicry of the beta-hairpin, because the hairpin is structured, but the C termini of tubulin much less. If the authors want to propose this, crosslinking the motor to the beta-hairpin or titration experiments varying the amount of hairpin added to motors in the presence of microtubules would support this (as for Kinesin-1 and Kinesin-13). They could also do experiments in the presence of microtubules or mutate the motor at the sites identified in figure 5. Alternatively, editing the text to remove the molecular mimicry message would work.

We think there has been a misunderstanding here, which has prompted us to clarify our language. We did not mean that the β -hairpin motif mimics the disordered C-terminal tails of tubulin. Rather, we meant that the residues in the Kif3AB motor domains involved in the β -hairpin motif and C-terminal coiled coil interaction are the same as those used to bind the globular core of tubulin. In this sense, the negative charged surface of the β -hairpin motif and C-terminal coiled coil mimic the negatively charged surface of the tubulin globular core. We have edited the text, which we hope now avoids this ambiguity.

Minor points

1. Introduction. Could give more background on kinesin 2 function at the start. Currently this information is in the first paragraph of the discussion.

We have elaborated on kinesin-2 function in the Introduction.

2. Can you comment on the affinity of KAP for Kif3a/b apart from the nanomolar affinity? Are they obligate interactors in the cell?

Our new cellular localization experiments show that Kap3 localization closely matches that of Kif3B, in agreement with a tight interaction between them in the cell. However, it is not possible to say that Kif3AB never exists without Kap3 bound i.e. they are true obligate interactors. For example, as highlighted by Scholey (2013), when Kif3 is purified from cells, the molar ratio of the Kap3 subunit is usually estimated to be slightly less than one (e.g. 1:1:0.8 in the case of the KifA:Kif3B:Kap3 orthologs in *C. reinhardtii*) (PMID: 23750925).

3. What is the behavior of kif3c? Do we expect the same properties? Does it have different properties or expression in cells?

As shown in **Extended Data Fig. 3**, Kif3AC-Kap3 is predicted to adopt a similar autoinhibited architecture to Kif3AB-Kap3. Our single-molecule binding experiment

(**Extended Data Fig. 3d**) shows that Kif3AC binds to Kap3 in a tail-dependent manner, similar to Kif3AB. Interestingly, however, while Kif3AC is predicted to use Interface 1 and 2 to bind Kap3, in a similar fashion to Kif3AB, Kif3AC is predicted to lack Interface 3 with Kap3 (**Extended Data Fig. 3c**). *In vivo*, Kif3AC's role is different to Kif3AB, as it does not function in intraflagellar transport and is restricted to cytoplasmic transport. Thus, Interface 3, which involves the C-terminal region of Kif3B contributing a β -strand and α -helix to a β -sheet at the Kap3 N-terminus, may dictate Kif3AB specificity for intraflagellar transport.

4. Page 5. First sentence, it would be good to say the mutations are in the context of the full-length motor.

Amended.

5. Figure 2. The diagrams are slightly confusing because the experiments are on the construct containing the motor domain. In figure 2c, they could show a close up on the full-length motor with the hairpin, with exactly the same schematics.

Thank you for this suggestion. We have modified the schematic to make it clear that the tail diagrams represent a closeup of a larger construct.

6. When structures generated by AlphaFold are presented, the authors should make it clear in the figure panel. Because there are also some figures with EM data, it can lead the reader to believe the structural models are derived from EM data.

We have emphasized in the figure legends which panels show AlphaFold3 models and which show EM data. We also now write 'AlphaFold3' in full, rather than using the 'AF3' abbreviation, to further clarify this point.

Reviewer #3 (Remarks to the Author):

Webb and coauthors employ a host of approaches (biochemistry, in vitro motility assays, immunofluorescence of fixed cells, and negative-stain electron microscopy) to characterize the molecular mechanism of kinesin-2 autoinhibition. They show the beta-hairpin motif in the tail of kinesin-2s mediates autoinhibition. Further, autoinhibition can occur when the kinesin-2 adaptor KAP3 is bound, where KAP3 instead forms an interface for interactions with other cargo adaptors.

Kinesin-2s drive intraflagellar transport and axonal transport of vesicular cargoes and mRNA. While the autoinhibition mechanism of kinesin-1 has been elucidated in ever greater detail, relatively little was known about how kinesin-2's activity is regulated. This manuscript makes an important contribution by filling this gap. Experiments were performed carefully, analyzed with rigor, and interpreted appropriately.

Despite my enthusiasm for the manuscript, there are several points that could be strengthened:

- The electron microscopy and alpha fold predictions indicate an autoinhibited KIF3AB-Kap3 structure, with the motor domains in close proximity to the beta-hairpins. Is there also evidence of an active, unfolded conformation in any of the class averages? What fraction of particles are represented in the class averages presented?

Thank you for raising this interesting point. To address this and Reviewer #2 point 2, we used negative stain EM to investigate the architecture of Kif3AB-Kap3 with APC^{ARM} bound, which we show is activated in single-molecule motility assays. As shown in **Extended Data Fig. 4d**, Kif3AB-Kap3-APC^{ARM} often has an extended appearance (**Extended Data Fig. 3b**, arrowheads). In contrast, Kif3AB-Kap3 in the absence of APC^{ARM} is predominantly in a compact conformation (>70% of particles) (micrograph shown in **Extended Data Fig. 3b**, representative class averages shown in **Fig. 4b**). This is in agreement with the strong autoinhibition of Kif3AB-Kap3 alone in single-molecule assays. We would prefer not to put a precise number on the fraction of molecules in the compact conformation from the EM experiment, because the nature of the remaining particles is unclear – they are not obviously extended (unlike the case with APC) and could represent compact particles that are less well stained and resolved.

- While the data make it clear that the beta-hairpin is sufficient to autoinhibit kinesin-2, do the authors expect other interactions between the coiled-coil regions of kinesin-2, similar to recent results for kinesin-1 (Cianfracco and McKenney labs), contribute to stabilizing the autoinhibited conformation?

As detailed in **Extended Data Fig. 5**, our studies suggest that the coiled coil conformation in auto-inhibited kinesin-2 is strikingly different to kinesin-1. Whereas in kinesin-1, the coiled coil adopts a hierarchical folding pattern, in which segments of the coiled coil pack against each other but the two α -helices remain in close apposition, for kinesin-2, our data suggest that the two α -helices separate from one another and pack against themselves to shield their hydrophobic seams in the auto-inhibited state, effectively opening up the coiled coil (**Extended Data Fig. 5a**). This self-packing of the separated α -helices is likely to stabilize the autoinhibited confirmation of kinesin-2. We have added a comparison to kinesin-1 to the Discussion. Thank you for this suggestion.

- Are there specific interactions between the beta-hairpins and APC that would exclude autoinhibition when bound to an adaptor? Can these data provide insight into cases where kinesin-

2 is expected to be bound to a cargo adaptor but in an autoinhibited state, for example in IFT and melanosome transport?

Yes, APC^{ARM} interacts with the Kif3A β -hairpin (**Fig. 6c**) and its position sterically clashes with the Kif3A motor domain in the autoinhibited conformation (**Fig. 6d**), providing a basis for why APC^{ARM} relieves auto-inhibition. Our model suggests that APC^{ARM} would need to at least partially disengage in order for kinesin-2 to resume the autoinhibited state. Whether the cargo adaptors used in IFT and melanosome transport occupy the same footprint as APC^{ARM} and the show the same behavior is a fascinating question for future studies.

- The descriptions of methods are overly reliant on references to previous work. Please include adequate details to assess the results (passivation, kymograph analysis).

Thank you for flagging this. We have added full descriptions to the Methods section.

- The run lengths for the deltaC156 construct are ~50% lower than reported in Guzik-Lendrum et al., 2015, Biophys. J.), although the results are consistent with Andreasson et al., 2015, Current Biology. There is a large variance in the run lengths for the deltaC156 construct between the three experiments. Can your experiments provide any insight into the cause of these discrepant results between labs and the variance observed in these results?

We think that the majority of the run length variance is explained by the assay buffer differences (particularly ionic strength) and the different constructs used in previous studies (Guzik-Lendrum *et al.* focused on constructs containing the motor domain, neck linker, and $\alpha 7$ helix, C-terminally fused in-register to the dimerization motif from EB1 or a synthetic heterodimerization domain; constructs in Andreasson *et al.* and this study used the native coiled coil stalk).

- Run lengths are similar to the resolution of diffraction-limited images. What is the expected resolution of the analysis used here?

- Analysis of the distributions of run lengths and velocities might provide insight into the co-existence of activated/inhibited states at different conditions / truncations.

We did not observe evidence for multimodal velocity distributions or multiphasic run length distributions. This is in-line with our proposed mechanism of auto-inhibition: when the Kif3 motor domains engage the β -hairpin motifs, microtubule binding is sterically blocked, and the motor is dissociated from its track (rather than being able to bind the microtubule and move slowly, which would be detectable as a subpopulation in the distributions). The pixel size of the TIRF microscope is 89 nm. We agree that the run lengths are close to the resolution of the analysis. That said, they are in good agreement with the optical trapping study of the equivalent construct by Andreasson *et al.* (2015), which is not diffraction limited.

- In Fig. 3, the effects of the different KIF3B constructs may result from variable expression levels of the constructs. These results would be strengthened by quantification of the protein levels of the KIF3B constructs in the cells compared. Do the KIF3B mutant constructs localize differently as would be expected?

We have performed this experiment, and apologize for not referring to it with sufficient clarity in the original submission. As shown in **Extended Data Fig. 2d**, the expression level is comparable for the different Kif3B constructs. The Kif3B KKK mutant accumulates strongly at the the ciliary tip whereas the Kif3B WT construct is localized relatively evenly along the length of the cilium (**Fig. 3b**). In response to this comment and Reviewer #2 point 1, we

generated new stable cell lines to examine the localization of Kap3. We found that Kap3 also accumulates at the ciliary tip in the background of the Kif3B KKK mutant (**Extended Data Fig. 2e**).

Minor comments:

- *What is the rationale for using different buffers for the APC experiments than the other assays?* We based this buffer on a previous study of APC and kinesin-2 (PMID: 32201729).

- *Lack of statistical comparisons (Fig. 2, 4G)* Amended.

- *Include a map of mutations in the APC-mut construct (Fig. 6)* Amended.

- *List the number of kymographs used for analysis (Fig. 2C)* We re-worked how the number of observations is reported in the Fig. 2 legend in response to this comment and Reviewer #1 point 2.

- *It would be instructive to include quantification of the mutant constructs in Fig. 2D*
Quantification in **Extended Data Table 1**.

Reviewer #4 (Remarks to the Author):

In their manuscript, Webb and coauthors use a combination of approaches (EM, Alphafold, single and multimolecule motility assays, biochemical analysis, cell biology) to identify the structural basis of the auto inhibition mechanisms of kinesin-2s. The results are novel and, in my mind very important. Kinesin-2's are key motor proteins the on and off switching of which is of utmost relevance to many cellular processes. Data and interpretation are extremely well presented and very convincing. The manuscript is very well written, also highly accessible to a broader audience. The manuscript fits well in the context of research performed on this topic before and the previous literature is well referenced.

I only have one key question to the authors. If that one is answered satisfactorily I think this manuscript should be published in NSMB, I think it is one of the best and most exciting manuscripts I have read this year!

My question is regarding the acidic residues in the KIF3A beta-hairpin. In figure 1A they are E615 and D616, while in the context of the autoinhibition (figure 5B, p9 l8) they are indicated as E612 and D613. Why are these numbers different? I hope this a simple miss labeling!

Thank you for this excellent catch. As well as the canonical Kif3A isoform in UniProt (entry Q9Y496), there is an isoform that has an additional 3 amino acids (J3KPF9), which is the origin of the numbering difference. We have unified the numbering to the canonical isoform.

In their revised version, the authors might consider the following textual suggestions (small points):
- p6 l11/12 I think it should be line (without 'S') scans. Amended.

- p6 l15 "The Kif3B beta-hairpin mutant" it would have helped me if you would have indicated also as "KKK" mutant (to connect with before and figure). Amended.

- in the discussion I was hoping for a short comparison of the kinesin-1 and kinesin-2 mechanisms and the reason / consequence why they are different. We have added a brief comparison with kinesin-1 to the Discussion. Thank you for this suggestion.

Erwin J.G. Peterman